# AlphaPeptDeep: a modular deep learning framework to predict peptide properties for proteomics

Wen-Feng Zeng [1], Xie-Xuan Zhou[1], Sander Willems[1], Constantin Ammar[1], Maria Wahle[1], Isabell Bludau [1], Eugenia Voytik[1], Maximillian T. Strauss [2] & Matthias Mann [1,2] ✉

Machine learning and in particular deep learning (DL) are increasingly important in mass spectrometry (MS)-based proteomics. Recent DL models can predict the retention time, ion mobility and fragment intensities of a peptide just from the amino acid sequence with good accuracy. However, DL is a very rapidly developing field with new neural network architectures frequently appearing, which are challenging to incorporate for proteomics researchers. Here we introduce AlphaPeptDeep, a modular Python framework built on the PyTorch DL library that learns and predicts the properties of peptides (https://github.com/MannLabs/alphapeptdeep). It features a model shop that enables non-specialists to create models in just a few lines of code. AlphaPeptDeep represents post-translational modifications in a generic manner, even if only the chemical composition is known. Extensive use of transfer learning obviates the need for large data sets to refine models for particular experimental conditions. The AlphaPeptDeep models for predicting retention time, collisional cross sections and fragment intensities are at least on par with existing tools. Additional sequence-based properties can also be predicted by AlphaPeptDeep, as demonstrated with a HLA peptide prediction model to improve HLA peptide identification for data-independent acquisition (https://github.com/MannLabs/PeptDeep-HLA).

The aim of MS-based proteomics is to obtain an unbiased view of the identity and quantity of all the proteins in a given system[1,2]. This challenging analytical task requires advanced liquid chromatography-mass spectrometry (LC/MS) systems as well as equally sophisticated bioinformatic analysis pipelines[3]. Over the last decade, machine learning (ML) and in particular deep neural network (NN)-based deep learning (DL) approaches have become very powerful and are increasingly beneficial in MS-based proteomics[4,5].

Identification in proteomics entails the matching of fragmentation spectra (MS2) and other properties to a set of peptides. Bioinformatics can now predict peptide properties for any given amino acid sequences so that they can be compared to actual measured data. This can markedly increase the statistical confidence in peptide identifications.

To do this, a suitable ML/DL model needs to be chosen which is then trained on the experimental data. There are a number of peptide properties that can be predicted from the sequence and for each of them different models may be most appropriate. For the peptide retention times in LC, relatively straightforward approaches such as iRT-calculator, RTPredict, and ELUDE have shown good results[6–8]. However, large volumes of training data are readily available in public repositories today and DL models currently tend to perform best[9]. This

[1]Department of Proteomics and Signal Transduction, Max Planck Institute of Biochemistry, Martinsried, Germany. [2]Proteomics Program, NNF Center for Protein Research, Faculty of Health Sciences, University of Copenhagen, Copenhagen, Denmark. ✉e-mail: mmann@biochem.mpg.de

is also the case for predicting the fragment intensities in the MS2 spectra, where DL models such as DeepMass:Prism[10], Prosit[11], our previous model pDeep[12,13], and many subsequent ones now represent the state-of-the-art. They mostly use long-short term memory (LSTM[14]) or gated recurrent unit (GRU[15]) models. Recently, transformers have been adopted in proteomics and show better performance[16,17]. This illustrates the rapid pace of advance in DL and the need for updating proteomics analysis pipelines with them. However, the focus of existing efforts has not been on extensibility or modularity, making it difficult or in some cases impossible to change or extend their NN architectures.

Here we set out to address this limitation by creating a comprehensive and easy to use framework, termed AlphaPeptDeep. As part of the AlphaPept ecosystem[18], we keep its principles of open source, community orientation as well as robustness and high performance. Apart from Python and its scientific stack, we decided to use PyTorch,[19] one of the most popular DL libraries.

AlphaPeptDeep contains pre-trained models for predicting MS2 intensities, retention time (RT), and collisional cross sections (CCS) of arbitrary peptide sequences or entire proteomes. It also handles peptides containing post-translational modifications (PTMs), including unknown ones with user-specified chemical compositions. AlphaPeptDeep makes extensive use of transfer learning, drastically reducing the amount of training data required.

In this paper, we describe the design and use of AlphaPeptDeep and we benchmark its performance for predicting MS2 intensities, RT, and CCS on peptides with or without PTMs. On challenging samples like HLA peptides, AlphaPeptDeep coupled with its built-in Percolator[20] implementation dramatically boosts performance of peptide identification for data-dependent acquisition. We also describe how AlphaPeptDeep can easily be applied to build and train models for different peptide properties such as a model for human leukocyte antigen (HLA) peptide prediction, which narrows the database search space for data-independent acquisition, and hence improves the identification of HLA peptides with the AlphaPeptDeep-predicted spectral library.

## Results

### AlphaPeptDeep overview and model training

For any given set of peptide properties that depend on their sequences, the goal of the AlphaPeptDeep framework is to enable easy building and training of deep learning (DL) models, that achieve high performance given sufficient training data (Fig. 1a). Although modern DL libraries are more straightforward to use than before, designing a neural network (NN) or developing a deployable DL model for proteomics studies is not as simple as it could be, even for biologists with programming experience. This is because of the required domain knowledge and the complexity of the different steps involved in building a DL model. The framework of AlphaPeptDeep is designed to address these issues (Fig. 1b).

The first challenge is the embedding, which maps amino acid sequences and their associated PTMs into a numeric tensor space that the NN needs as an input. For each amino acid, a 'one-hot encoder' is customarily used to convert it into a 27-length fingerprint vector consisting of 0 s and 1 s (Methods). In contrast, PTM embedding is not as simple. Although recent studies also used one-hot encoding to embed phosphorylation for MS2 prediction via three additional amino acids[16], this is not extendable to arbitrary PTMs. In pDeep2 (ref. 13), the numbers of C, H, N, O, S, P atoms for a site-specific modification are prepended to the embedding vector which is flexible and can be applied to many different PTMs. AlphaPeptDeep inherits this feature from pDeep2 but adds the ability to embed all the other chemical elements. To make the input space manageable, we use a linear NN that reduces the size of the input vector for each PTM (Methods, Supplementary Fig. 1). This allows efficient embedding for most modification

types, except for very complex ones such as glycans. The PTM embedding can be called directly from AlphaPeptDeep building blocks.

To build a new model, AlphaPeptDeep provides modular application programming interfaces (APIs) to use different NN architectures. Common ones like LSTM, convolutional NN (CNN) as well as many others are readily available from the underlying PyTorch library. Recently transformers – attention-based architectures to handle long sequences – have achieved breakthrough results in language processing but were then also found to be applicable to many other areas like image analysis[21], gene expression[22] and protein folding[23]. AlphaPeptDeep includes a state-of-the-art HuggingFace transformer library[24]. Our framework also easily allows combining different NN architectures for different prediction tasks.

The training and transfer learning steps are mostly generic tasks, even for different NNs. Therefore, we designed a universal training interface allowing users to train the models using just a single line of Python code – 'model.train()'. In our training interface, we also provide a "warmup" training strategy to schedule the learning rate for different training epochs (Methods). This has proven very useful in different tasks to reduce the bias at the early training stage[25]. Almost all DL tasks

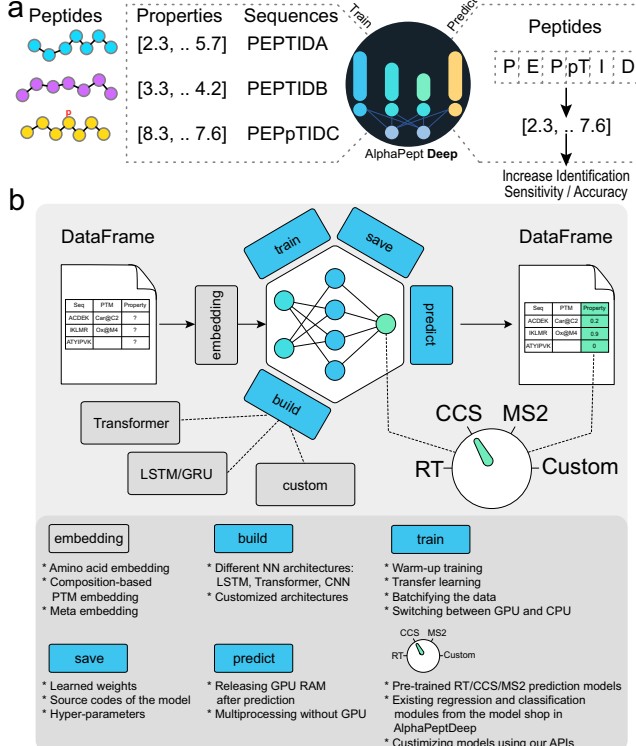

**Fig. 1 | Overview of the AlphaPeptDeep framework. a** Measured peptide properties are encoded with the respective amino acid sequences and used to train a network in AlphaPeptDeep (left). Once a model is trained, it can be used on arbitrary sets of peptide sequences to predict the property of interest. This can then improve the sensitivity and accuracy of peptide identification. **b** The AlphaPeptDeep framework reads and embeds the peptide sequences of interest. Its components include the build functionality in which the model can build. Meta embedding refers to the embedding of meta information such as precursor charge states, collisional energies, instrument types, and other non-sequential inputs. It is then trained, saved and used to predict the property of interest. The dial represents the different standard properties that can be predicted (RT retention time, CCS collision cross section, MS2 intensities of fragment spectra). Custom refers to any other peptide property of interest. The lower part lists aspects of the functionalities in more detail. NN neural network, LSTM long short-term memory, CNN convolutional neural network, GRU gated recurrent unit, API application programming interface, PTM post-translational modification.

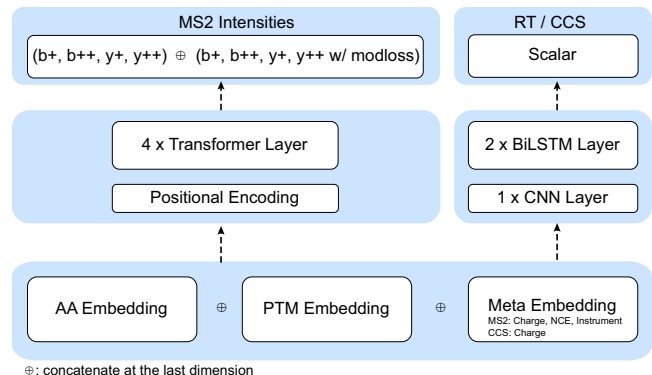

**Fig. 2 | The built-in and pre-trained MS2, RT, and CCS prediction models.** The MS2 model is built on four transformer layers, and the RT/CCS models consist of a convolutional neural network (CNN) layer followed by two bidirectional long short-term memory (BiLSTM) layers. The pre-trained MS2 model currently supports predicting the intensities of backbone b/y ions as well as their modification-associated neutral losses if any (e.g. −98 Da loss of phosphorylation on Ser/Thr). However, the user can easily configure the MS2 model to train and predict water and ammonium losses from backbone fragments as well. RT retention time, CCS collision cross section, MS2 intensities of fragment spectra, BiLSTM bidirectional long short-term memory, CNN convolutional neural network, AA amino acid, PTM post-translational modification, NCE normalized collision energy.

can be done on graphical processing units (GPUs) and training a model from scratch on a standard GPU usually take not more than hours in AlphaPeptDeep and is performed only once. Transfer learning from a pre-trained model is feasible within minutes, even without GPU.

After training, all learned NN parameters should be saved for persistent applications. This can be readily done using DL library functionalities, and is also implemented in AlphaPeptDeep – 'model.save()'. In the latter case, AlphaPeptDeep will save the source code of the NN architectures in addition to the training hyperparameters. Thus, the NN code and the whole training snapshot can be recovered even if the source code was accidentally changed in the AlphaPeptDeep or developers' codebase. This is especially useful for dynamic computational graph-based DL libraries such as PyTorch and TensorFlow in 'eager mode' because they allow dynamically changing the NN architectures.

The most essential functionality of the AlphaPeptDeep framework is the prediction of a property of a given peptide of interest. When using only the CPU, one can choose multiprocessing (predicting with multiple CPU cores), making the prediction speed acceptable on regular personal computers (PCs) and laptops (nearly 2 h for the entire reviewed human proteome). On our datasets and hardware, prediction on GPU was about an order of magnitude faster. As PyTorch caches the GPU RAM in the first prediction batch, subsequent batches for the same model will be even faster. However, GPU random access memory (RAM) should be released after the prediction stage, thus making the RAM available for other DL models. These steps are automatically done in AlphaPeptDeep within the 'model.predict()' functionality.

AlphaPeptDeep provides several model templates based on transformers and LSTM architectures in the "*model shop*" module to develop new DL models and also allows choosing hyperparameters from scratch for classification or regression with very little code. All these high-level functionalities in AlphaPeptDeep give the user a quick on-ramp and they minimize the effort needed to build, train and use the models. As an illustrative example, we built a classifier to predict if a peptide elutes in the first or second half of the LC gradient using only several lines of code. Training took only ~16 min on nearly 350 K peptide-spectrum matches (PSMs) on a standard *HeLa* dataset[26] and the model achieved 95% accuracy in the testing set (Supplementary Fig. 2).

The MS2, RT, and CCS prediction models (Fig. 2) are released on our GitHub repository and will be automatically imported into AlphaPeptDeep when using the package for the first time. The MS2 prediction model was inherited from pDeep2 but reimplemented on transformers which have been shown very useful in MS2 prediction[16,17]. The pre-trained MS2 model in AlphaPeptDeep is much smaller than other models without sacrificing accuracy (4 M parameters vs 64 M in the Prosit-Transformer[17]), making the prediction very fast (Supplementary Fig. 3). Testing by the same 1.4 M peptides on the same GPU workstation showed that fragment intensity prediction of Alpha-PeptDeep is 40 times faster than Prosit-Transformer (35 s vs 24 min, Supplementary Fig. 3). We also applied the same principle of lightweight models to our RT and CCS models (less than 1 M parameters for each, Methods), which we built on previous LSTM models[26–28].

We trained and tested the MS2 models with ~40 million spectra from a variety of instruments, collision energies and peptides, and trained the RT and CCS models with about half a million RT and CCS values of peptides (Supplementary Data 1). The results of this initial training were then stored as pre-trained models for further use or as a basis for refinement with transfer learning.

Using these pre-trained models and specifically designed data structures (Methods), the prediction of a spectral library with MS2 intensities, RT, and ion mobilities (converted from CCS, Methods) for the human proteome with 2.6 M peptides and 7.9 M precursors took only 10 min on a regular GPU and 100 min on the CPU with multiprocessing (Supplementary Fig. 3). As this prediction only needs to be done at most once per project, we conclude that the prediction of libraries by DL is not a limitation in data analysis workflows.

## Prediction performance of the AlphaPeptDeep model for MS2 spectra

With the AlphaPeptDeep framework for prediction of MS2 intensities, RT and CCS in hand, we first benchmarked the MS2 model against datasets of tryptic peptides (phase 1 in Fig. 3a). The training and testing data were collected from various instruments and collisional energies, including ProteomeTools[29], which were derived from synthetic peptides with known ground truth. (Check Supplementary Data 1 for the detailed information of datasets.) We split the data sets in two and trained on a LSTM model similar to pDeep or on the new transformer model. As expected, transformer performed better than the LSTM model on the test datasets (Supplementary Fig. 4). Overall, on ProteomeTools data measured with different collisional energies on the Lumos mass spectrometer, 97% of all significantly matching PSMs had Pearson correlation coefficients (PCC) of the predicted vs. the measured fragment intensities of at least 90% (Fig. 3a), which we term 'PCC90' in this manuscript. Note that the experimental replicates also exhibit some variation, making the best possible prediction accuracy somewhat less than 100%. For example, on the ProteomeTools replicates generated from the Lumos, 99% had PCCs above 90% (Supplementary Data 1), meaning that our predicted intensities mirrored the measured ones almost within experimental uncertainties (99% experimental vs. 97% predicted). Next, we tested the model on the same ProteomeTools sample but measured on a trapped ion mobility Time of Flight mass spectrometer (timsTOF) in dda-PASEF mode[26,30], and achieved a PCC90 of 87.9% (Supplementary Data 1), showing that the prediction from the pre-trained model is already very good for timsTOF even without adaption.

As expected, our pretrained model performed equally well across different organisms, as demonstrated by PXD019086-Drosophila and -Ecoli in Fig. 3a. Interestingly, it did almost as well on chymotrypsin or GluC-digested peptides although it had not been trained on them (PXD004452-Chymo and -GluC in Fig. 3a).

HLA class 1 peptides are short pieces of cellular proteins (about 9 amino acids) that are presented to the immune system at the cell surface, which is of great interest to biomedicine[31]. Because of their low

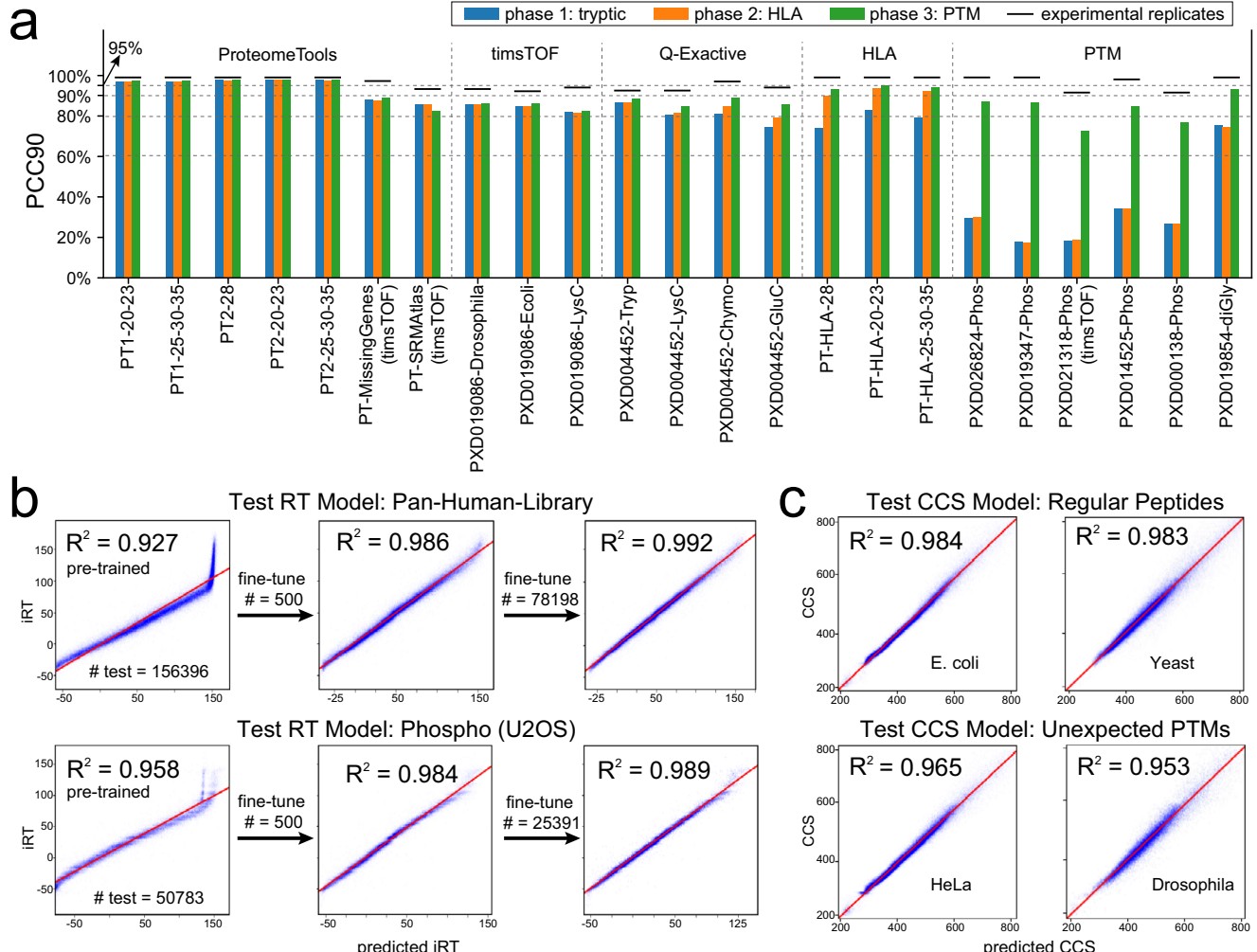

**Fig. 3 | The performance of MS2/RT/CCS models. a** The MS2 prediction accuracies of the three training phases on different testing datasets. The dataset names are on the x-axis. The performance is evaluated by "PCC90" (percentage of PCC values larger than 0.9). The prefix 'PT' of each data set refers to ProteomeTools. PT1 and PT2 refer to ProteomeTools part I and II, respectively. PT1, PT2, and PT-HLA were all measured with Lumos by the Kuster lab. The black bars are the PCC90 values of experimental spectra. **b** For RT prediction, few-shot learning can correct the RT bias between different LC conditions. Top panels: pan-human-library; bottom panels: phosphorylation dataset from U2OS. **c** Our CCS model works well for both regular (top panels, *E. coli* and *Yeast*) and unexpectedly modified (bottom panels, *HeLa* and *Drosophila*) peptides. RT retention time, CCS collision cross section, MS2 intensities of fragment spectra, PCC Pearson correlation coefficient, HLA human leukocyte antigen, PTM post-translational modification.

abundance and non-tryptic nature, they are very challenging to identify by standard computational workflow, a task in which DL can help[32]. In a second training phase, we appended a synthetic HLA dataset, which was also from ProteomeTools[33], into the training set of phase 1 and trained the model for additional 20 epochs ('fine tuning the model'). We first checked if the new model negatively impacted performance on the tryptic data sets, but this turned out not to be the case (phase 2 in Fig. 3a). On the HLA peptides, however, performance substantially increased the PCC90 from 79% to 92%.

Finally, we extended our model to predict phosphorylated and ubiquitylated peptides, which have spectra somewhat distinct from unmodified peptides. In this case, in addition to backbone fragmentation intensities, AlphaPeptDeep also needs to learn the intensities of fragments with or without modifications. For phosphopeptide prediction, performance of the pre-trained model was much worse, with PCC90 values of only around 30%. However, after training on PTM datasets at phase 3, the performance dramatically increased, almost to the level of tryptic peptides (Fig. 3a). The ubiquitylation prediction (rightmost in Fig. 3a) was already reasonable with the pre-trained model but increased further after phase 3 training (PCC90 from 75% to

93%). The final model was saved as the default pre-trained model in the AlphaPeptDeep package.

**Prediction performance of the AlphaPeptDeep models for RT and CCS**

RT and CCS models are quite similar to each other as their inputs are the peptide sequences and PTMs, and outputs are scalar values. For both we used LSTM architectures. In the CCS prediction model, precursor charge states are considered in the model as well. Taking advantage of the PTM embedding in AlphaPeptDeep, the RT and CCS models naturally consider PTM information, and hence can predict peptide properties given arbitrary PTMs. We trained the RT model on datasets with regular peptides from our *HeLa* measurements[26]. All the training and testing dataset information are listed in Table 1. 'Regular peptides' refers to unmodified peptides or modified peptides containing only Oxidation@M, Carbamidomethyl@C and Acetyl@Protein N-term.

We first tested the trained RT model on regular peptides from the PHL dataset. As shown in Fig. 3b, the pre-trained model gave very good predictions in most of the RT range, but failed to accurately predict the

**Table 1 | Dataset information used to train and test RT/CCS models**

| Dataset | Search | Modifications | Usage | Description |
|---|---|---|---|---|
| **RT model** | | | | |
| *HeLa* | MaxQuant | regular | training | Trypsin and LysC *HeLa* peptides. ref. [26] |
| PHL | | regular | testing | Pan human library. ref. [57] |
| Phos-U2OS | Spectronaut | regular and phos | testing | Phosphopeptides of U2OS. ref. [58] |
| **CCS model** | | | | |
| *HeLa* | MaxQuant | regular | training | Same as *HeLa* in RT section |
| *E. coli* | MaxQuant | regular | testing | *E. coli* peptides. ref. [26] |
| *Yeast* | MaxQuant | regular | testing | *Yeast* peptides. ref. [26] |
| HeLa-Open | Open-pFind | all possible PTMs | testing | Same as *HeLa* in RT section. Only peptides with nonregular modifications were kept after open-search for testing |
| *Drosophila*-Open | Open-pFind | all possible PTMs | testing | *Drosophila* peptides. ref. [26]. Only peptides with nonregular modifications were kept after open-search for testing |

'regular' in the 'Modifications' column refers to unmodified, Oxidation@M, Carbamidomethyl@C and Acetyl@Protein N-term. The 'Search' column with 'Open-pFind' means that we re-analyzed the MS data with Open-pFind (Methods), and only peptides with nonregular modifications were kept for testing. Otherwise, the search results were downloaded from the original publications of the datasets. *RT* retention time, *CCS* collision cross section, *PTM* post-translational modification.

last few minutes (iRT (ref. 7) values higher than 100) possibly due to the different flushing settings of the LC in training and testing data. These differences could be addressed by fine-tuning the model with experiment-specific samples. Few-shot fine-tuning with only 500 training samples improved the accuracies of the RT prediction from an R2 of 0.927 to 0.986.

We also tested the RT model on the Phos-U2OS dataset, although the model had not been trained on such phosphorylation data. After fine-tuning on 500 peptides, the R2 increased from 0.958 to 0.984 (Fig. 3b). As RT behavior of peptides varies with the LC conditions in different experiments, we highly recommend fine-tuning whenever possible. It turns out that few-shot fine-tuning worked well to fit short LC conditions as well (Supplementary Figure 5). Finally, as expected, the more training peptides we used, the better the fine-tuning, and with many peptides our model reached R2 values up to 0.99 (Fig. 3b).

While the CCS model was trained on regular human peptides from the same *HeLa* dataset as RT model training, we tested the trained model on different types of data (Table 1). First, we tested on regular peptides but from non-human species. Here we used *E. coli* and *yeast* peptides from the same instrument in the same publication. For these regular peptides the CCS model achieved an $R^2 > 0.98$ of the predicted and detected CCS values. Second, we tested on peptides with different modifications. For modified peptides from *HeLa*-Open and *Drosophila*-Open datasets (Table 1), the R2 was 0.965 and 0.953, respectively, a prediction accuracy quite close to the one for regular peptides, even for unexpected modifications. The predicted CCS values can be converted to ion mobilities on the Bruker timsTOF using the Mason Schamp equation[34].

### Prediction performance for 21 PTMs with transfer learning

To further demonstrate the powerful and flexible support for PTMs in AlphaPeptDeep, we tested the pre-trained tryptic MS2 model (model of phase 1 in Fig. 3a) and RT model using the 21 PTMs, which were synthesized based on 200 template peptide sequences[35].

Interestingly, there is a group of modifications for which the prediction of MS2 spectra is as good as the values of unmodified peptides (Fig. 4a). These include Hydroxypro@P, Methyl@R, and Dimethyl@R for which the PCC90 was greater than 80%. This is presumably because these modifications do not change the overall fragmentation pattern much. In contrast, most of the other PTMs cannot be well predicted by the pre-trained models, for example, the PCC90 values were less than 10% for Malonyl@K and Citrullin@R. We applied transfer learning for each PTM type using 10 or 50 training peptides with different charge states and collisional energies, reserving the remaining ones with the same PTM for testing of our transfer learned

models. Furthermore, we also trained with 80% of the peptides and tested on the remaining 20% (Fig. 4a). Remarkably, transfer learning on as few as ten peptides greatly improved the prediction accuracies on the testing data. The largest improvements of PCC90 were as high as 60% (Citrullin@R and Malonyl@K, Fig. 4a). Overall, compared with the pre-trained model, the ones tuned by 10 peptides improved the PCC90 from a median of 48% to 87% (Fig. 4b). We speculate that this is because the fragmentation properties of amino acids at different collisional energies have been well learned by the pre-trained model after which transfer learning only needs to learn the properties of modified ones. Including 50 PTM bearing peptides improved this number to 93% whereas using 80% of all the identified peptides ($n \le 200$) with these PTMs only improved prediction by another 2%. This demonstrates that our models can be adapted to novel situations with very little additional data, due to the power of transfer learning.

AlphaPeptDeep has been included in AlphaViz[36], a tool suite for RAW MS data visualization (https://github.com/MannLabs/alphaviz), which among other features allows users to visualize a mirrored plot between experimental and predicted spectra. As an example, the MS2 prediction of the peptide "AGPNASIISLKSDK-Biotin@K11" before and after transfer learning is displayed in Fig. 4c. The y12 ++ ion was first wrongly predicted by the pre-trained model, but this was corrected after transfer learning with only 50 other biotinylated peptides. We also generated mirrored MS2 plots for ten randomly selected peptides of each PTM type before and after transfer learning, see Supplementary Data 3. AlphaPeptDeep also allows users to visualize the 'attention' weights– a key feature of transformer models – showing what data attributes were important for the prediction. To depict the attention changes between pre-trained and transfer learning transformer models, we used the BertViz package (https://github.com/jessevig/bertviz) (Supplementary Fig. 6).

Next, we tested the performance of our pre-trained RT model using the datasets of 21 PTMs. Although the model was never trained on any of these PTMs, the accuracy of RT prediction on these peptides exceeds that of DeepLC[37], an RT prediction model designed for unseen PTMs (R2 of 0.95 of AlphaPeptDeep vs. 0.89 of DeepLC, Figs. 4d and 4e). In this case, transfer learning only slightly improves the results, presumably because some of these synthetic modified peptides elute in very broad peaks, which makes them hard to predict.

### Boosting data-dependent acquisition (DDA) identification of HLA peptides

As explained above, HLA peptides are among the most challenging samples for MS-based proteomics. Given the excellent model performance of the transformers in AlphaPeptDeep, we hypothesized that

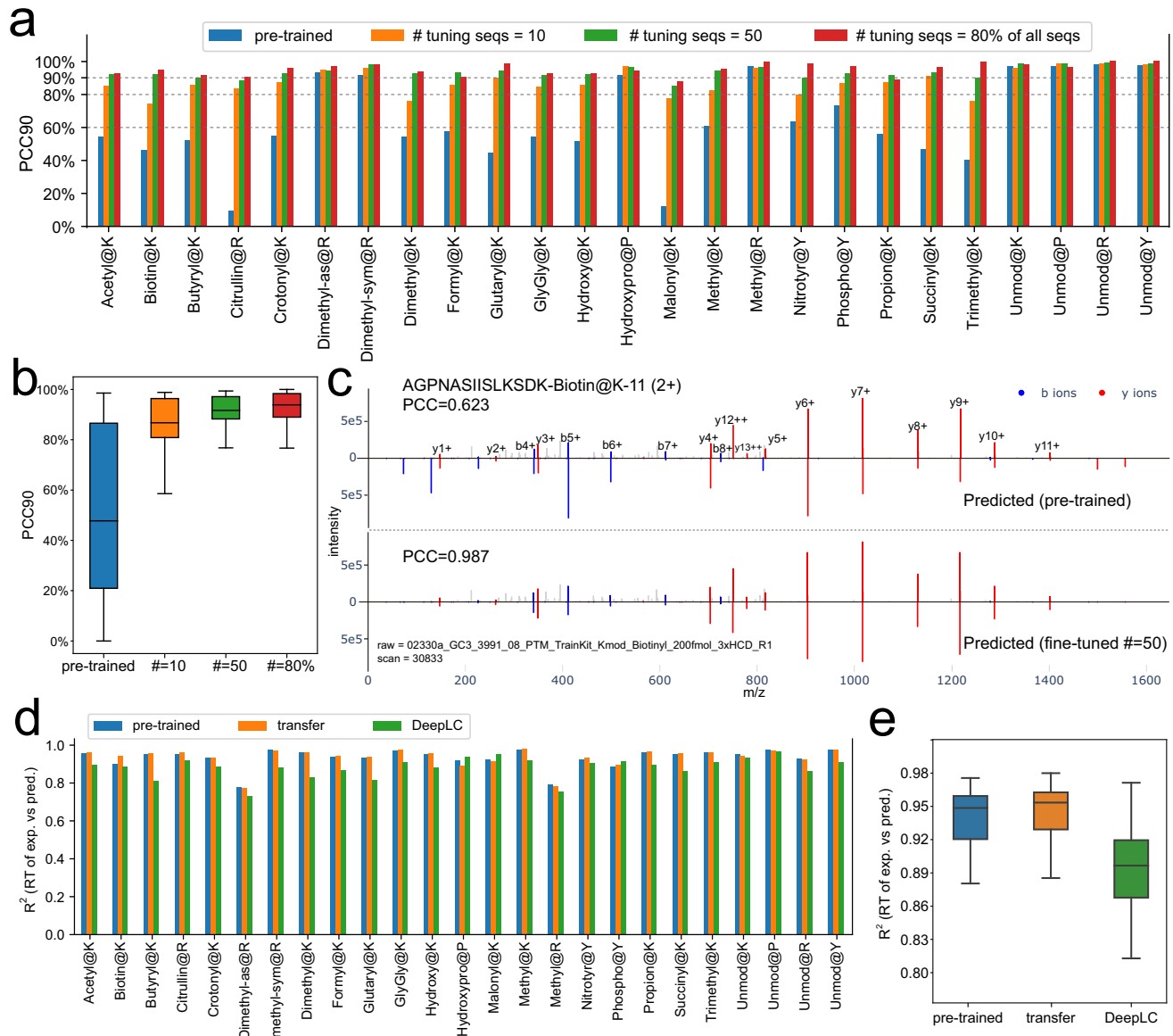

**Fig. 4 | Model performance with transfer learning on 21 PTMs from Proteome Tools. a** The accuracy of MS2 prediction with different numbers of peptides for transfer learning for each PTM. Each PTM is tested separately. "80% seqs" refers to using 80% of the identified modified sequences for transfer learning. **b** Overall accuracy without unmodified peptides from (**a**). Boxplots for $n = 21$ PCC90 values of 21 PTM types are drawn with the interquartile range within the boxes and the median as a horizontal line. The whiskers extend to 1.5 times the size of the interquartile range. **c** Transfer learning dramatically improves the MS2 prediction of the example peptide "AGPNASIISLKSDK-Biotin@K11" (tuned by 50 other peptides). **d** Comparisons of RT prediction for each PTM on pre-trained and transfer learning (by 50% of all the identified peptides) models, as well as DeepLC models. **e** Overall $R^2$ distribution without unmodified peptides from (**d**). Boxplots for $n = 21$ $R^2$ values of 21 PTM types are drawn with the interquartile range within the boxes and the median as a horizontal line. The whiskers extend to 1.5 times the size of the interquartile range. RT retention time, MS2 intensities of fragment spectra, PCC Pearson correlation coefficient, PTM post-translational modification.

prediction of their MS2 spectra could substantially improve their identification.

The non-tryptic nature of these peptides results in an very large number of peptides that need to be searched, leading to a decreased statistical sensitivity at a given false discovery rate (FDR) level (usually 1%). The key idea of using MS2, RT and CCS prediction to support HLA peptide identification is that, for correct peptides of the searched spectra, the predicted properties should be very close to the detected ones, while the predicted properties of the irrelevant peptides tend to be randomly distributed. Therefore, the similarities or differences between the predicted and detected properties can be used as machine learning features to distinguish correct from false identifications using semi-supervised learning. Such an approach has been

implemented in tools coupled with Percolator[20] to re-score PSMs, which increases the sensitivity at the same FDR level[32,33]. But due to the lack of support for arbitrary PTMs with DL models this has not been implemented for open-search. However, AlphaPeptDeep now is able to predict the properties of arbitrarily modified peptides, and even HLA peptides with unexpected PTMs. This feature is intended to boost the identification of HLA peptides in conjunction with modern open-search engines like pFind[38], which identify unexpected PTMs by using the sequence tag technique[39].

AlphaPeptDeep applies the semi-supervised Percolator algorithm[40] on the output of the search engines, rescoring PSMs to better discriminate true identifications from false ones based on deep learning predicted parameters (Methods) Rescoring for the

open-search is also supported. To accelerate the rescoring, we calculate the fragment intensity similarities between predicted and detected spectra on a GPU, making the rescoring process very fast. On our PC with a GeForce RTX 3080 GPU, it took ~1 h to rescore 16,812,335 PSMs from 424 MS runs, where ~60% of the time was used for loading the RAW files. Running without the GPU on the same PC took ~3.5 h, whereas non-specific open-search for this many spectra took more than a week, meaning that the rescoring by AlphaPeptDeep is not a bottleneck for HLA peptide search.

To investigate how much AlphaPeptDeep can boost the HLA peptide search, we applied it on two datasets, MSV000084172 containing 424 RAW files from samples in which particular mono-allelic HLA-I types were enriched[41], here referred to as the 'mono-allelic dataset' and our published dataset from tumor samples (PXD004894[42] with 138 RAW files) referred to as the 'tumor dataset'. These two datasets had been analyzed with a regular search engine (MaxQuant[43]) by the Kuster group[33] (Fig. 5a) and we here used pFind in the open-search mode (Fig. 5b).

First, we wanted to compare the AlphaPeptDeep results with MaxQuant as well as Prosit, a recently published DL based tool that has also been applied to HLA peptides[33]. The MaxQuant PSMs and Prosit-rescored PSMs were downloaded from ref. 33 and the MaxQuant PSMs were rescored by AlphaPeptDeep for comparison. Since Prosit only supports fixed iodoacetamide modification on alkylated peptides (IAA in Fig. 5a), we only used the results of the same IAA RAW files in rescoring. On the mono-allelic and the tumor datasets, AlphaPeptDeep covered 93% and 96% of the MaxQuant results while more than doubling the overall numbers at the same FDR of 1% (Fig. 5a). Compared to Prosit, AlphaPeptDeep captured 91% of their peptides and still improved the overall number on the mono-allelic dataset by 7%.

Next, we searched both datasets with the open-search mode of pFind (Fig. 6b), and rescored the results in AlphaPeptDeep. Here, both alkylated and non-alkylated peptides were analyzed. Interestingly, the open-search itself already identified similar numbers of peptides as the DL-boosted regular search, but AlphaPeptDeep further improved the total number of identified peptides by 29% and 42%, while retaining 99% and 98% of the pFind hits at the same FDR for the mono-allelic and tumor datasets, respectively (Fig. 5b). This demonstrates the benefits of AlphaPeptDeep's support of open-search for HLA peptide analysis.

AlphaPeptDeep with open-search identified PTMs such as phosphorylation, which are known to exist on HLA peptides but are very difficult to identify by regular unspecific search[44]. For the mono-allelic dataset we identified a total of 490 phosphopeptides. To gauge the biological reasonability of these peptides, we searched for sequence motifs of both the phosphorylated and non-phosphorylated peptides. This revealed the expected HLA peptide motifs, dominated by the anchor residues for their cognate major histocompatibility complex proteins. Only the phospho-HLA peptides additionally had linear phospho-motifs, like the prominent Ser-Pro motif common to proline directed kinases (Fig. 5c and Supplementary Fig. 7). We also identified 359 phospho-HLA peptides from the tumor dataset, with similar phospho-motifs (Supplementary Fig. 7). We further used AlphaPeptDeep to inspect retention time and MS2 spectrum similarities. The results demonstrated an 80% PCC90 of phospho-HLA PSMs which is close to unmodified ones, and RT differences from predicted to measured peptides were also close to zero (Supplementary Fig. 8). Furthermore, we manually validated 300 randomly selected HLA PSMs for these HLA peptides including 44 phosphopeptides using an extension to AlphaViz[36]. Their annotated and mirrored MS2 plots can be found in Supplementary Data 4. This independently verified our model and assignments. Note that the MS2 and RT models were only fine-tuned by at most 100 phospho-PSMs from eight RAW files (Methods), so most of the phosphopeptides from remaining RAW files (i.e., 416 out of 424 and 130 out of 138 RAW files in the mono-allelic and tumor dataset, respectively) were not used in fine-tuning. Our method

was also able to identify other PTMs associated with HLA peptides, such as cysteinylation[45] (Supplementary Fig. 9). Overall, most of the HLA peptides additionally identified by this method had modifications related to sample preparation, such as deamidation, N-terminal pyro-Glu, and N-terminal carbamidomethylation (Supplementary Fig. 9).

## Building an HLA prediction model for HLA DIA search

In recent years, DIA has become a method of choice to generate large-scale proteome datasets. DIA data analysis traditionally requires DDA experiments to generate a library to which the data is then matched[46]. These libraries contain RT, ion mobility (if applicable) and the most intense and specific fragments for each peptide. The generation of experimental libraries is laborious and sample consuming. With the development of DL in proteomics, libraries with predicted RT, CCS/ion mobilities and fragment intensities from whole proteome sequences are becoming more and more popular, although there is still a debate about whether measured or predicted libraries are preferable. This is because the large search space introduced by purely in silico libraries can make FDR control difficult.

DIA for HLA peptide analysis is also getting more attention[47,48]. So far, these efforts have been restricted to experimental DDA libraries because analysis with a predicted HLA library from proteome sequences is far more challenging than with an experimental one. This is mainly because HLA peptides are not tryptic, meaning they do not follow specific cleavage rules and do not necessarily have a favorable fragmentation pattern. The number of theoretical peptides with amino acid lengths between 8 and 14 from a reviewed human proteome is more than 70 M, which is nearly two orders of magnitude more than that of tryptic peptides in the same length range (~900 K). Due to this enormous search space, a predicted library is difficult or even impossible to search by state-of-the-art DIA search tools such as DIA-NN[49] and Spectronaut[50].

Fortunately, HLA peptides follow certain sequence motifs guided by the HLA-types that are present. We reasoned that these motifs could be learned by DL for more efficient peptide identification. To test this hypothesis, we built an HLA prediction model using the model shop functionalities in our AlphaPeptDeep framework (Methods). In this model a binary LSTM classifier predicts if a given sequence is likely to be an HLA peptide presented to the immune system and extracts these peptides from the human proteome sequence. There are two main goals of the model: (1) keep as many actually presented HLA peptides as possible (i.e., high sensitivity); and (2) reduce the number of predicted peptides to a reasonable number (i.e., high specificity). Note that sensitivity is more important here as we hope that all measured HLA peptides are still in the predicted set.

Based on these goals, we developed a pipeline which enables predicted library search for DIA data. In brief, we divided our pipeline into five steps, as shown in Fig. 6a. In step 1, we trained a pan-HLA prediction model with peptides from known HLA allele types ('pan-HLA model' in Fig. 6a). Normally, up to 6 different allele types are present in the samples from any given individual. Therefore, in step 2, we used transfer learning to create a person-specific model with sample-specific peptides identified from individuals ('sample-specific model' in Fig. 6a). This model should then be able to predict whether an HLA peptide is potentially present in the sample or not, thus further reducing the number of peptides to be searched and increasing prediction accuracy. For this strategy, we need to identify a number of sample specific HLA peptides. This can be done directly from the already acquired DIA data by a 'direct-DIA search'[51] obviating the need for a separate DDA experiment. This involves grouping eluting fragment detected peaks belonging to the same peptide signal into a pseudo-spectrum for DIA data, and then searching the pseudo-spectrum with conventional DDA search algorithms. In step 3, we used the sample-specific model to predict all possible personalized HLA peptides directly from the protein sequence database (i.e. the

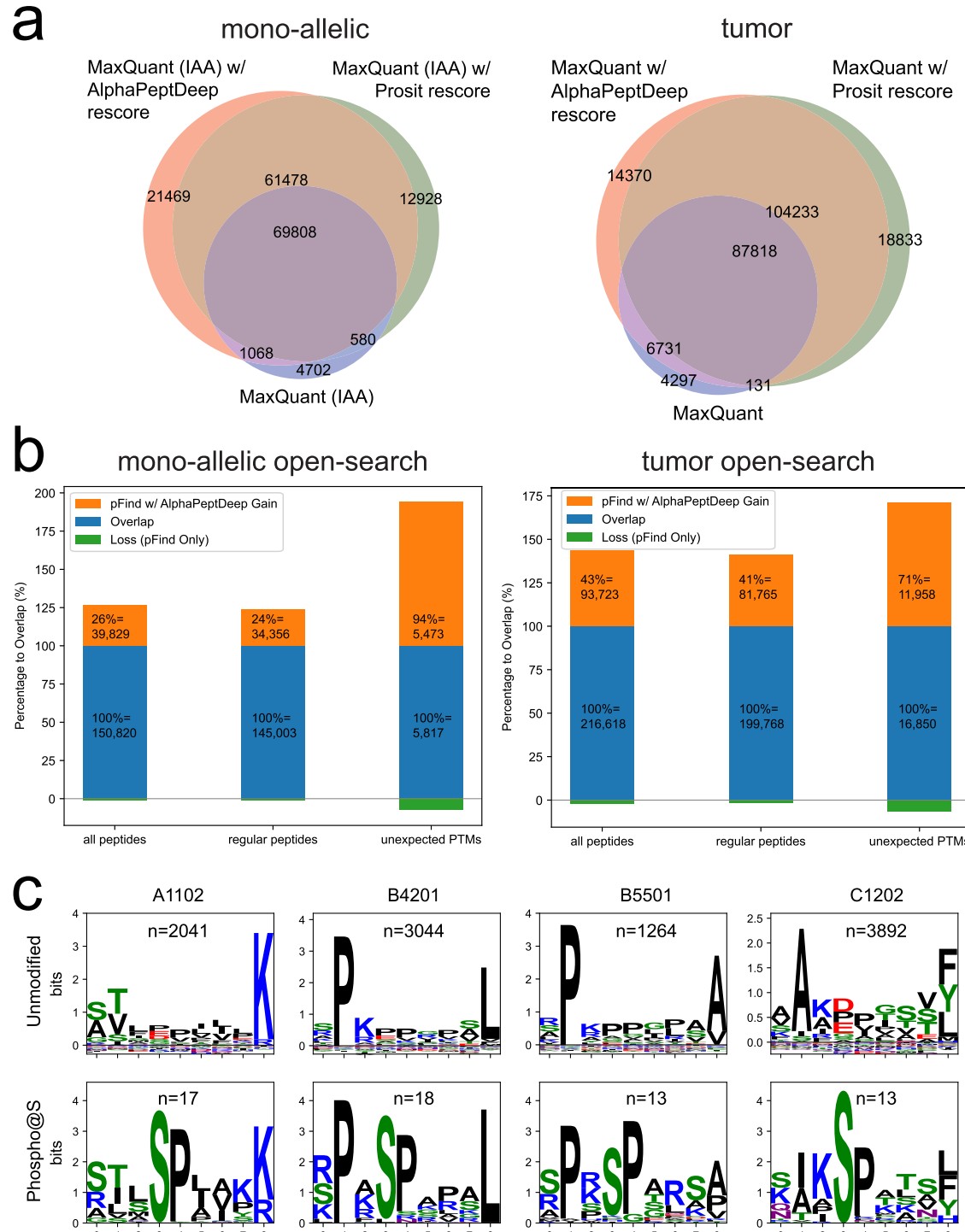

**Fig. 5 | AlphaPeptDeep drastically improves results for DDA identification of HLA peptides. a** Improvement upon regular search (MaxQuant). IAA refers iodoacetamide alkylated peptides. **b** Improvement upon open-search (pFind). Regular peptides here refer to peptides without modifications or those with only Met-oxidation and Cys-alkylation. **c** Logo plots of unmodified and phosphorylated peptides with nine amino acids identified by open-search for four different HLA allele types. Logo plots were generated by LogoMaker.[59] HLA human leukocyte antigen, DDA data-dependent acquisition.

fasta file). These predicted HLA peptides were used to generate a predicted spectral library by using AlphaPeptDeep (step 4) and were then identified by DIA data with DIA search engines.

To test this pipeline, we used the HLA-I dataset of the RA957 cell line in PXD022950[47]. We started with our pan-HLA prediction model trained by 80% of the peptides and tested on the remaining 20% from the 94 known HLA allele types (Fig. 5). This reduced the number of sequences from 70 M to 7 M with 82% sensitivity on the testing set.

However, 7 M peptides are still too many to search and the model would have lost 18% of true HLA peptides. Furthermore, the pre-trained model is not able to identify unknown HLA allele types as it is only trained on already known ones.

To enable transfer learning, we searched RA957 data with DIA-Umpire[51]. It identified 12,998 unique sequences with length from 8 to 14. We used transfer learning on 80% of this data to train the sample specific HLA model while keeping 20% for testing. This dramatically

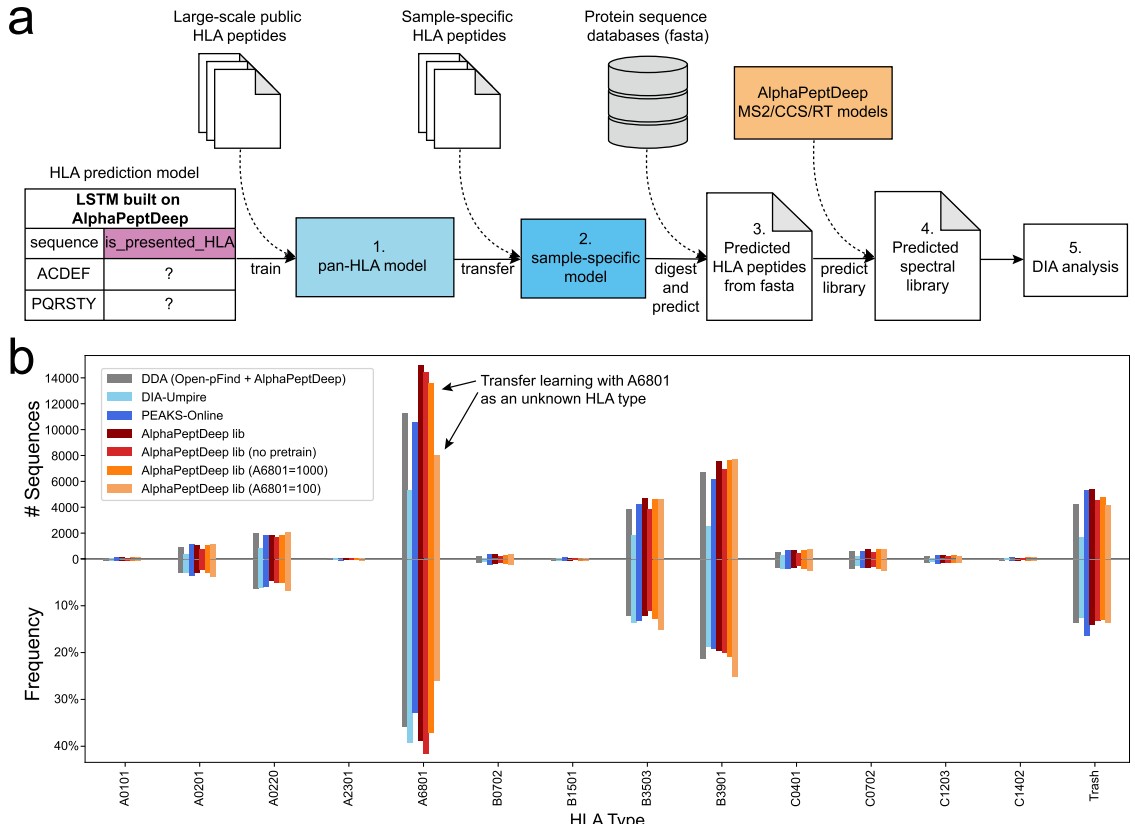

**Fig. 6 | HLA prediction model built on AlphaPeptDeep functionalities. a** The pipeline with the HLA prediction model to extract potential HLA peptides from the proteome databases. The HLA model is a binary classifier that predicts if a given sequence is a potentially presented HLA sequence. **b** Our HLA prediction model boosts the number of identified HLA-I peptides compared to other tools. Cell line HLA data from RA957 with sequence lengths from 8 to 14 were used. The top bar plots show the number of identified unique sequences of HLA allele types for each search method. The bottom bar plots the relative frequency of these HLA allele types. 'Trash' means the peptides cannot be assigned to any HLA allele types by MixMHCpred at 5% rank level. 'AlphaPeptDeep lib' (red) refers to the library predicted by the sample-specific HLA model and our MS2 and RT models. The bars represent DDA data analyzed by MaxQuant, and the DIA data analyzed by DIA-Umpire, or PEAKS-Online including de novo sequencing. AlphaPeptDeep with transfer learning for the sample-specific HLA library clearly outperforms these. Training on the sample-specific peptides without transfer learning obtained similar number of identified HLA peptides ("no pretrain" in (**b**)). The results of omitting the dominant HLA-A*68:01 (A6801) HLA type in the pan-model and using transfer learning with including 1000 or 100 of these peptides identified by direct-DIA from the data are shown in the last two bars of the A6801 type (see arrows in the panel). RT retention time, CCS collision cross section, MS2 intensities of fragment spectra, HLA human leukocyte antigen, DDA data-dependent acquisition, DIA data-independent acquisition.

increased the specificity to 96% with 92% sensitivity (note that this is judged on the identifications by direct-DIA; thus our sensitivity may be even higher). The number of HLA peptides predicted by this model is 3 M, which is more comparable to the tryptic human proteome library.

Having predicted our sample-specific HLA peptides, including their MS2 fragment spectra and RTs, we used this as input for a DIA-NN search of the DIA data. Our workflow identified 36,947 unique sequences. PEAKS-Online[52] is a very recently published tool which combines searching a public library, direct-DIA, and de novo sequencing. It identified 30,733 unique sequences within the same length range. Our workflow almost tripled the number of unique sequences of DIA-Umpire and obtained 20% more than PEAKS-Online. As a reference, the DDA search on the data in the original publication[47] but using Open-pFind rescored by AlphaPeptDeep identified 30,039 unique sequences in the 8 to 14 aa range.

To judge the reliability of the identified HLA peptides, we used MixMHCpred[53] to deconvolute these identified peptides at the 5% rank level based on the HLA type list in the original publication of the datasets[47] (Fig. 6b). Our pipeline identified more unique HLA sequences than the best DDA workflow and other DIA workflows. Additionally, the overall peptide distribution identified by our pipeline for different HLA allele types was very similar to that of the DDA data, indicating that our additionally identified HLA peptides were reliable at the same

level. We also manually checked randomly selected 100 peptides by plotting their elution profiles and mirrored MS2 annotations (Supplementary Data 5). The superiority of our workflow is possibly owing to our predicted HLA peptides that cover most of the present HLA peptides but exclude unlikely ones, making the search space much smaller and therefore avoiding false hits.

Interestingly, training directly on the direct-DIA peptides without transfer learning resulted in only slightly fewer peptides than transfer learning ("no pretrain" in Fig. 6b). As there are not many motifs for each individual, we reasoned that it should be straightforward to learn the sequential patterns from only thousands of HLA peptides. This would be useful to identify unknown sample-specific HLA allele types as we do not need any prior knowledge. Note that transfer learning is still necessary if we only have small number of training peptides and unknown allele types. To simulate this situation, we removed all peptides of HLA-A*68:01 from the 94 allele types, and used the rest to train a new pan-HLA model. This means that all HLA-A*68:01 peptides in the RA957 sample were now unknown. Then we used only 100 HLA-A*68:01 and all non-HLA-A*68:01 peptides identified by direct-DIA and deconvoluted by MixMHCpred for transfer learning. The resulting library then identified 29,331 peptides including 7,868 from HLA-A*68:01 (Transfer learning with 1000 HLA-A*68:01 peptides retrieved almost all of them) (Fig. 6b). This demonstrates that few-shot transfer

learning is able to rescue many of the peptides of an unknown HLA type even if the peptide number is low after direct-DIA identification.

## Discussion

We developed a deep learning framework called AlphaPeptDeep that unifies high-level functionalities to train, transfer learn and use models for peptide property prediction. Based on these functionalities, we built MS2, RT and CCS models, which enabled the prediction for a large variety of different PTM types. These models can boost DDA identification of for example, HLA peptides, not only in regular search but also in open-search. We also provided a module called 'model shop' which contains generic models so that users can develop new ones from scratch with just a few lines of code. Based on the model shop, we also built an HLA prediction model to predict whether a peptide sequence is a presented HLA peptide. With the HLA model and the MS2, RT and CCS models in AlphaPeptDeep, we predicted the HLA spectral libraries directly from the whole human proteome, and searched them using HLA DIA data. Using our predicted libraries outperformed existing DDA and DIA workflows. However, this does not prove that DIA is always better than DDA in HLA peptidome analysis, as the DDA proteome database is 20 times larger than our predicted library for DIA analysis. Future DDA search engines may be able to identify more peptides if they supported predicted library search.

Although AlphaPeptDeep is both powerful and easy to use, we note that traditional machine learning issues, such as overfitting in the framework, still need to be kept in mind. For instance, users still need to split the data, train and test the models on different sets. Trying different hyperparameters such as the number of training epochs is still necessary as well. Different mini-batch sizes and learning rates may also impact on the model training. However, the model shop at least provides baseline models for any property prediction problem.

We hope AlphaPeptDeep will minimize the challenges for researchers that are not AI experts to build their own models either from scratch or on top of our pre-trained models. As we pointed out in our recent review[4], peptide property prediction can be involved in almost all steps to improve the computational proteomics workflow. Apart from specific properties of interest in MS-based proteomics, it can in principle be used to solve any problem where a peptide property is a function of the amino acid sequence, as we demonstrated by successfully predicting potential HLA peptides to narrow the database search. Therefore, with sufficient and reliable training data, we believe AlphaPeptDeep will be a valuable DL resource for proteomics.

## Methods

### Infrastructure development

To develop AlphaPeptDeep, we built an infrastructure package named AlphaBase (https://github.com/MannLabs/alphabase) which contains many necessary functionalities for proteins, peptides, PTMs, and spectral libraries. In AlphaBase, we use the pandas DataFrame as the base data structure, which allows transparent data processing in a tabular format and is compatible with many other Python packages. AlphaPeptDeep uses the AlphaBase DataFrames as the input to build models and predicts properties of peptides. Amino acid and PTM embedding is performed directly from 'sequence' (amino acid sequence), 'mods' (modification names), and 'mod_sites' (modification sites) columns in the peptide DataFrame.

AlphaBase uses UniMod modification names to represent modifications, and designates the modified amino acids by "@", e.g Oxidation@M, Acetyl@Protein N-term, etc. It provides result readers to import DDA and DIA search engines (e.g., AlphaPept[18], MaxQuant[43], DIA-NN[49], Spectronaut), and translates common modification names (e.g., oxidation, acetylation, carbamidomethylation, phosphorylation, di-gly) into AlphaBase format. Users can also provide modification dictionaries to tell AlphaBase how to translate other modifications for search engines. As pFind already uses UniMod names, AlphaBase supports all of pFind modifications, thus parsing open-search results is straightforward.

### Amino acid embedding

Each amino acid of a sequence is converted to a unique integer, for example, 1 for 'A', 2 for 'B', ..., and 26 for 'Z'. Zero is used as a padding value for N- and C-terminals, and other "padding" positions. As a result, there are 27 unique integers to represent an amino acid sequence. A 'one-hot encoder' is used to map each integer into a 27-D vector with zeros and ones. These vectors are mapped to an N-dimensional embedded vector using a linear layer (Supplementary Figure 1). For this, we additionally make use of the 'torch.Embedding' method, which is more efficient and flexible and can support more letters such as all the 128 ASCII codes.

### PTM embedding

For each PTM, we use a 6-D embedding vector to represent the C, H, N, O, S, and P atoms. All other atoms of a PTM are embedded into a 2-D vector with a fully connected (FC) layer. The 6-D and 2-D vectors are concatenated into an 8-D vector to represent the PTM (Supplementary Fig. 1).

### MS2 model

The MS2 model consists of an embedding layer, positional encoder layer, and four transformer layers followed by two FC layers. The embedding layer embeds not only amino acid sequences and modifications but also metadata (if necessary) including charge states, normalized collisional energies, and instrument type. All these embedded tensors are concatenated for the following layer.

We added an additional transformer layer to predict the 'modloss', which refers to neutral loss intensities of PTMs, for example, the $-98\,Da$ of the phospho-group. This modloss layer can be turned off by setting 'mask_modloss' as 'True'. The output layer dimension is ($n - 1) \times 8$ for each peptide, where $n$ is the length of the peptide sequence, and 8 refers to eight fragment types, i.e., b+, b++, y+, y++, b_modloss+, b_modloss++, y_modloss+, and y_modloss++. With 'mask_modloss=True', the modloss layer is disabled and the predicted modloss intensities are always zero. The hidden layer size of transformers is 256. The total number of the model parameters is 3,988,974.

All matched b/y fragment intensities in the training and testing datasets were normalized by dividing by the highest matched intensity for each spectrum. The MS2 models were trained based on these normalized intensities. For prediction, negative values will be clipped to zero, hence the predicted values will be between zero and one.

In training phase 1, we only used tryptic peptides in the training datasets. The training parameters were: epoch=100, warmup epoch=20, learning rate (lr)=1e−5, dropout=0.1. In training phase 2, HLA peptides were added to the training set and the parameters were: epoch=20, warmup epoch=5, lr=1e−5, dropout=0.1, mini-batch size=256. In phase 3, phosphorylation and ubiquitylation datasets were added for training, and only phosphorylation sites with >0.75 localized probabilities were considered. The training parameters were: epoch=20, warmup epoch=5, lr=1e−5, dropout=0.1, mini-batch size=256. For transfer learning of the 21 PTMs, the parameters were: epoch=10, warmup epoch=5, lr=1e−5, dropout=0.1, mini-batch size depends on the peptide length. L1 loss was used for all training phases. We used the "cosine schedule with warmup" method implemented in HuggingFace for warmup training of these models including all the following models.

For Thermo Orbitrap instruments, the fragment intensities of each identified PSM are directly extracted from the raw data. For this, we imported the centroided MS2 spectra with Thermo's RawFileReader API that is integrated in AlphaPept, hence the extracted intensities are reproducible across different search engines. For dda-PASEF data, the b/y ion intensities are extracted directly from the

msms.txt file of MaxQuant results. Note that different search engines may have different centroiding algorithms for dda-PASEF, resulting in quite different fragment intensities, so fine-tuning is highly recommended for dda-PASEF data analyzed by different software.

A fragment DataFrame is designed to store the predicted intensities. Its columns are fragment ion types (e.g., 'b_z1' for b+ and 'y_z2' for y ++ ions), and the rows refer to the different fragmented positions of peptides from which the fragments originate. The start and end pointers of the rows ('frag_start_idx' and 'frag_end_idx') belonging to peptides are stored in the peptide DataFrame to connect between peptides and their fragments. The fragment DataFrame is pre-allocated only once for all peptides before prediction. While predicting, the predicted values of a peptide are assigned to the region of the peptide located by 'frag_start_idx' and 'frag_end_idx'. The fragment DataFrame allows fast creation and storage of the predicted intensities. The tabular format further increases human readability and enables straightforward access by programming.

### RT model
The RT model consists of an embedding layer for sequences and modifications, and a CNN layer followed by two LSTM layers with a hidden layer size of 128. The outputs of the last LSTM layer are summed over the peptide length dimension and processed by two FC layers with output sizes of 64 and 1. The total number of the model parameters is 708,224.

All RT values of PSMs in the training datasets were normalized by dividing by the time length of each LC gradient, resulting in normalized RT values ranging from 0 to 1. As a result, the predicted RTs are also normalized. The training parameters were: epoch=300, warmup epoch=30, lr=1e−4, dropout=0.1, mini-batch size=256. The fine-tuning parameters are: epoch=30, warmup epoch=10, lr=1e−4, dropout=0.1, mini-batch size=256. L1 loss was used for training.

To compare predicted RT values with experimental ones, each value is multiplied with the time length of each LC gradient. For testing on peptides with iRT values, we used 11 peptides with known iRT values[7] to build a linear model between their iRT and predicted RT values. Then all the predicted RTs in the testing sets are converted to iRT values using the linear model.

### CCS model
The CCS model consists of an embedding layer for sequence, modifications and charge states, and a CNN layer followed by two LSTM layers with a hidden layer size of 128. The outputs of the last LSTM layer are summed over the peptide length dimension and processed by two FC layers with output sizes 64 and 1. The total number of the model parameters is 713,452.

The training parameters are: epoch=300, warmup epoch=30, lr=1e−4, dropout=0.1, mini-batch size=256. L1 loss was used for training. The predicted CCS values are converted to mobilities of Bruker timsTOF using the Mason Schamp equation[34].

### Rescoring
Rescoring includes three steps:
1. Model fine-tuning. To reduce overfitting, only 5,000 PSMs are randomly sampled from at most eight RAW files at 1% original FDR reported by the search engine to fine-tune the MS2, RT and CCS (if applicable) models to obtain project-specific models. The top-10 frequent modifications are also selected for fine-tuning from the eight RAW files. At most 100 PSMs are sampled for each modification. Therefore, the fine-tuning covers not only unmodified peptides, but also modified ones.
2. Deep learning feature extraction. The tuned MS2, RT and CCS models are used to predict MS2, RT and CCS values for all the reported PSMs including decoys. All PSMs are matched against the MS2 spectra in the RAW files to obtain detected fragment

intensities. Then the predicted and detected values are used to calculate 61 score features, which include correlations of fragments, RT differences, mobility differences, and so on (Supplementary Data 2).
3. Percolator for rescoring. We use the cross-validation schema[54] to perform the semi-supervised Percolator algorithm to reduce the chance of overfitting. All the peptides are divided into K folds (K = 2 in the analyses of this work) and rescored by 5 iterations in Percolator. In each iteration, a Logistic regression model from scikit-learn[40] is trained with the 61 features on the K−1 folds, and the model is used to re-score on the remainder. All the K folds will be re-scored after repeating this for K times on each of the folds.

Multiprocessing is used in step 2 for faster rescoring. Because GPU RAM is often limited, it can become a bottleneck meaning that only one process is allowed to access the GPU space at a time for prediction. We developed a producer-consumer schema to schedule the tasks with different processes (Supplementary Fig. 10). The PSMs are matched against MS2 spectra in parallel with multiprocessing grouped by RAW files. Then, they are sent back to the main process for prediction in GPU. At last, the 61 Percolator features are extracted in parallel again. All correlation values between matched and predicted MS2 intensities are also calculated in GPU for acceleration; as this is not memory intensive, the GPU RAM can be shared and used in parallel from different processes. For multiprocessing without GPU, all predictions are done with separate processes and results are merged into the main process to run Percolator.

### HLA prediction model
The HLA prediction model consists of an embedding layer for sequences, a CNN layer followed by two LSTM layers with a hidden layer size of 256. The outputs of the last LSTM layer are summed over the sequence length dimension and processed by two linear layers with output sizes of 64 and 1. The sigmoid activation function is applied for last linear layer to obtain probabilities. The total number of the model parameters is 1,669,697.

For training and transfer learning, identified HLA peptides with sequence lengths from 8 to 14 are regarded as positive samples. Negative samples were randomly picked from the human protein sequences with the same length distribution as the HLA peptides. These samples were then split 80% for training and 20% for testing. The parameters for training the pre-trained model were: epoch=100, warmup epoch=20, lr=1e−4, dropout=0.1. For transfer learning, the DIA data were searched by DIA-Umpire and MSFragger[55] in HLA mode at 1% FDR with reviewed human protein sequence. The parameters for transfer learning were: epoch=50, warmup epoch=20, lr=1e−5, dropout=0.1, mini-batch size=256. Binary cross-entropy loss was used for training.

To predict HLA peptides from fasta files, we first concatenate protein sequences into a long string separated by the "$" symbol. Next, we use the longest common prefix (LCP) algorithm[56] to accelerate the unspecific digestion for the concatenated sequence. Only the start and end indices of the peptides in concatenated sequence are saved, thus minimizing the usage of RAM. These indices are used to generate peptide sequences on the fly for prediction. The LCP functionalities have been implemented in AlphaBase. All sequences with a predicted probability larger than 0.7 were regarded as potential HLA peptides.

### Open-search for Orbitrap and dda-PASEF data
We performed an open search on the Thermo RAW data with OpenpFind. For HLA DDA data, the reviewed human protein sequences from UniProt (https://www.uniprot.org/) were searched with the following parameters: open-search mode=True, enzyme=Z at C-terminal (i.e., unspecific enzyme), specificity=unspecific. The search tolerance was set to ±10 ppm for MS1 and ±20 ppm for MS2. All modifications

marked as 'isotopic label' in UniMod (www.unimod.org) were removed from the searched modification list. The FDR was set as 1% at the peptide level.

To enable Open-pFind search for dda-PASEF data, the spectra were loaded by AlphaPept APIs[18] and exported as pFind compatible MGF files using our in-house Python script. The reviewed drosophila and human sequences were used to search the respective tryptic DDA data with parameters: open-search mode=True, enzyme=KR at C-terminal, enzyme specificity=specific. The search tolerance was set to ±30 ppm for both MS1 and MS2.

## Spectral libraries

Functionalities for spectral libraries are implemented in AlphaBase. When providing DataFrames with sequence, modification and charge columns, the fragment m/z values and intensities are calculated and stored in fragment DataFrames. AlphaBase also integrates functionalities to load and save DataFrames in a single Hierarchical Data Format (HDF) file for fast access. For subsequent use with DIA-NN or Spectronaut, all the DataFrames are then converted into a tab-separated values file (*.tsv) which is compatible with these tools.

For HLA DIA analysis, we used reviewed human protein sequences to predict HLA peptides. We considered charge states from one to three for each peptide. All RT, CCS, and MS2 were predicted using the model from training phase 3. The 12 most abundant b/y ions with 1+ and 2+ charge states were written to the *.tsv file. Fragment m/z range was set to be from 200 to 1800, precursor m/z range was from 300 to 1800.

In DIA-NN, the mass tolerance for MS1 and MS2 were set to 10 and 20 ppm respectively, with a scan window of 8. All other parameters were the default values of DIA-NN. The results identified from the first pass were used for post-search analysis.

## Reporting summary

Further information on research design is available in the Nature Portfolio Reporting Summary linked to this article.

## Data availability

The reviewed protein sequence databases are downloaded from uniprot: https://www.uniprot.org/proteomes/UP000005640 for human, https://www.uniprot.org/proteomes/UP000000625 for *E. coli*, https://www.uniprot.org/proteomes/UP000001744 for *fission yeast*, and https://www.uniprot.org/proteomes/UP000000803 for *drosophila*.

The training and testing data were from ProteomeXchange with accession codes: PXD010595, PXD004732, PXD021013, PXD009449, PXD000138, PXD019854, PXD019086, PXD004452, PXD014525, PXD017476, PXD019347, PXD021318, PXD026805, PXD026824, PXD029545, PXD000269, and PXD001250.

The mono-allelic HLA DDA dataset was downloaded from MassIVE with accession code MSV000084172.

The tumor HLA dataset was downloaded from ProteomeXchange with accession code PXD004894.

HLA DIA data and the MaxQuant results of DDA data from the RA957 cell line were downloaded from PRIDE with accession code PXD022950. HLA DIA results of PEAKS-Online were taken from the Supplementary Data files in[52]. Only results from RAW files '20200317_QE_HFX2_LC3_DIA_RA957_R01.raw' and '20200317_QE_HFX2_LC3_DIA_RA957_R02.raw' from RA957 were used to compare different methods.

Source data files and Python notebooks for data analysis in this study are provided on https://doi.org/10.6084/m9.figshare.20260761.

## Code availability

The source code of AlphaBase and AlphaPeptDeep are fully opened on GitHub: https://github.com/MannLabs/alphabase and https://github.com/MannLabs/alphapeptdeep. They are also available through PyPI

with "pip install alphabase" and "pip install peptdeep". The versions of AlphaBase and AlphaPeptDeep used in this study are 0.1.2 and 0.1.2 respectively. All the pre-trained MS2, RT, and CCS models can be found in https://github.com/MannLabs/alphapeptdeep/releases/download/pre-trained-models/pretrained_models.zip. These models will be automatically downloaded when using the AlphaPeptDeep package for the first time.

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

## Acknowledgements

We thank Marvin Thielert for the testing of the spectral libraries. We thank Mario Oroshi and Igor Paron for help with retrieval of MS RAW data. This study was supported by The Max-Planck Society for the Advancement of Science and by the Bavarian State Ministry of Health and Care through the research project DigiMed Bayern (www.digimed-bayern.de). I.B. acknowledges funding support from her Postdoc.Mobility fellowship granted by the Swiss National Science Foundation [P400PB_191046]. MTS is supported financially by the Novo Nordisk Foundation (Grant agreement NNF14CC0001). M.W. and I.B. are both supported financially by European Union's Horizon 2020 research and innovation programme (grant agreement No 874839 ISLET).

## Author contributions

W.-F.Z. developed AlphaBase, AlphaPeptDeep and models, and analyzed the data. X.-X.Z. developed the models, contributed to AlphaPeptDeep code, and analyzed the data. S.W. designed the template code structure on GitHub and developed HDF functionalities in AlphaBase. C.A. reviewed the code and contributed to the model shop functionalities. M.W. came up with the idea of HLA prediction. I.B. contributed to AlphaBase. E.V. developed functionalities in AlphaViz for mirrored MS2 plots and testing the integration with AlphaPeptDeep. M.T.S. reviewed almost all the source code of AlphaBase and AlphaPeptDeep, and provided a lot of suggestions. M.M. supervised this project. W.-F.Z., M.T.S. and M.M. wrote the manuscript. All the authors revised the manuscript.

## Funding

## Competing interests

The authors declare no competing interests.
