## [Peer Review File · Nature Communications]

REVIEWER COMMENTS

Reviewer #1 (Remarks to the Author):

The manuscript presents a modular deep learning framework to predict the LC-MS-related properties of a given peptide sequence including retention time, ion mobility, and fragment pattern. This framework does not only provide users with high-performance pre-trained models, but also the capability of deploying new neural network architectures for prediction tasks. It may generate project-specific models via transfer learning, which improves the performance greatly in a variety of applications such as HLA peptide identification from DDA and DIA data. Overall, I think it is a valuable framework for exploring deep learning solutions in MS-based proteomics.

What I concern is about the validity of those additionally identified HLP peptides. As the ground truth is lacking, it is hard to know they are true or false positives. Hence, it might be a good idea to have several case studies presented here, for instance randomly select a small number of HLP peptides, mirror-plot their experimental and predicted spectra, and/or draw their precursor and fragment ion elution profiles in case of DIA data. Such manual inspection shall reveal how reliable those identifications are to some extent.

Reviewer #2 (Remarks to the Author):

The authors present a Python package, AlphaPeptDeep. The package is meant to make the use of deep learning methods in MS-based proteomics more accessible to non-experts. The package also comes equipped with a set of pre-trained models for MS2 fragmentation, retention time, and CSS prediction, all of which can be fine-tuned for a user's specific dataset. The package is evaluated in several experiments, with special attention given to immunopeptidomics MS experiments.

I am impressed with the idea behind AlphaPeptDeep and the authors' work in putting it together. However, I think that there are issues to be addressed in both the manuscript and the online documentation for the tool before it can be considered for publication.

In general, I think there needs to be more in the manuscript on how AlphaPeptDeep is used. A lot of the manuscript is evaluating trained models, but the actual thing that is new and novel here is the Python package that was used to create all those models. That is the exciting thing, that someone who knows some Python can install a single package and then build deep learning models for predicting pretty much any peptide property they can come up with so long as it is dependent upon peptide sequence. Obviously, we need to know that models created for standard tasks like MS2 prediction work as expected, but more emphasis should be given to the novel aspect here. The models themselves are not the novelty in this work (with the exception of the HLA library generation, which is quite neat and does showcase how new things can be done with AlphaPeptDeep).

I have worked a lot with different machine learning-based tools for proteomics, and user-friendliness has almost always been an issue. AlphaPeptDeep intends to provide an easy-to-use package for developing, training, fine-tuning, and running predictors for any peptide property which depends on peptide sequence. I think AlphaPeptDeep is on the right track to do this. The authors want to make an easy way to train arbitrary deep learning models for tasks in proteomics, which is a fantastic idea. They have really hit on the fact that predicting peptide properties is just a sequence-to-sequence or sequence-to-scalar task, and that many problems can be addressed with models having the same basic structure. However, I think there needs to be much more in the way of documentation and online tutorials for it to really be user-friendly and approachable for non-experts. This is an integral part of making useful software tools.

I have a lot of comments and questions on the manuscript, which will follow. I hope they are not taken harshly, because I think the tool has a lot of potential. However, I think work needs to be done to clarify some parts of the manuscript and the use of the tool. First, however, I will discuss my experience using AlphaPeptDeep, which I think illustrates the need for more comprehensive documentation.

Please note that my comments below reflect the online documentation at 19:00 hr CET August 13, 2022. If changes were being made to the documentation on an ongoing basis, I might have missed those changes.

The manuscript points the reader to a GitHub page for AlphaPeptDeep. I followed this and found a nicely organized repository with some well-written instructions for installing the tool from the command line. I also saw that there is a GUI available which could be downloaded from the releases section. I am a bioinformatician and programmer, but being a human I am also somewhat lazy and so I thought the GUI would be the simplest thing to try out first. After downloading it I tried to build a predicted spectral library. I typically work with peptides and not whole proteins, so I tried using the peptide table input, but ran into the problem that there was no description of what the peptide table should look like. I looked on the GitHub repository, but again there was no explanation and no

examples. There is a section for modification information in the GUI, but it is more like what you expect in a search engine: a list of fixed and variable modifications. Since there is no description or examples of how to format a peptide table, I thought that perhaps it just takes a peptide list and then iterates through all the possible charge states and variable modifications. However, when I tried loading the peptide list, the GUI printed out an error message that it could not determine the file delimiter (though it is very nice that the GUI shows the error traceback). Obviously, it is expecting a delimited file (like as CSV or TSV), but I don't know what to give it and I can't find out from the available resources.

Since the peptide table input could not be used, I tried giving it a FASTA file with a few proteins in it. That worked, and very well! Within minutes I had a predicted spectral library, and it looked exactly how it should. That satisfied me that the package generally works as described, and that the problem is that there was simply no way to know what kind of input the tool was expecting for the peptide table (except for the FASTA).

Despite the fact that the GUI is prominently mentioned on the GitHub page, it was not mentioned in the manuscript. Thus, I assumed that the GUI is in development and that is why there are no built-in instructions and no online documentation. I next tried the CLI. It was easily installed into a clean Python virtual environment using the instructions in the GitHub repository (`pip install peptdeep`).

When I tried running the CLI, it started fine and showed the expected help screen:

```
'''
```

```
$ peptdeep -h
```

```
Usage: peptdeep [OPTIONS] COMMAND [ARGS]...
```

Options:

`-v, --version` Show the version and exit.

`-h, --help` Show this message and exit.

Commands:

`gui` Start graphical user interface.

`install-models` Install peptdeep pretrained models.

`library` Predict library for DIA search.

rescore Rescore DDA results.

transfer Transfer learning for different data types.

...

I next wanted to know how to generate a library, so I tried getting help on that function, but the CLI told me nothing helpful:

...

```
$ peptdeep library -h
```

```
Usage: peptdeep library [OPTIONS] SETTINGS_YAML
```

Predict library for DIA search.

Options:

```
-h, --help Show this message and exit.
```

...

It seems to expect a YAML file with some settings in it (as well as some options), but there are no example YAML files that I could easily find on the GitHub page, nor any documentation on what it should look like.

The same commands for “rescore” and “transfer” gave similar output. The only documentation in the GitHub repository for the CLI is this: “It is possible to get help about each function and their (required) parameters by using the -h flag.” There is no other documentation, tutorials, or examples.

If one is technically inclined, one could load AlphaPeptDeep in a running Python environment and use it from there. There is documentation for the API linked from the GitHub repository, but it is also lacking in examples or tutorials. The API documentation itself is pretty sparse, as there are no definitions given for any function parameters. Thus, if one wants to really know how to use the tool, they will need to go to the source code. I think very few people are going to go that far...

It seems, at least based on the current state of the documentation, that the tool is unusable by people who do not already know how to use it. The idea behind AlphaPeptDeep is great. It will be wonderful for people who have a little coding experience, but not enough to be comfortable diving into PyTorch or Tensorflow, to be able to develop and train a deep learning model for prediction of arbitrary peptide properties. But without proper documentation and tutorials to tell them how to do

it, there is no way for them to get started. In short, even as an expert I would not be able to easily use AlphaPeptDeep to reproduce the DL models produced by the authors. As that production is an important part of the results, I believe this issue needs to be addressed prior to any possible publication.

Having said all that, I have seen enough that I believe the tool does work, and that the authors have put a lot of time and effort into it. I hope they will put together some excellent documentation to show others how to use AlphaPeptDeep, because it does seem like something that could really benefit the proteomics community.

The remainder of my comments will be on the manuscript itself.

pp 2, lines 51-52 : “This is also the case for predicting the fragment intensities in the MS2 spectra, where DL models such as our previous model pDeep, DeepMass:Prism, Prosit and many subsequent ones now represent the state-of-the-art.”

This sentence makes it sound like all three of these tools might come from the same group, which they do not.

pp 2, lines 75-76 : “AlphaPeptDeep dramatically boosts performance of peptide identification for data-dependent acquisition”

This implies that AlphaPeptDeep is directly involved in the identification process. Most readers will probably understand what you are getting at, but at this point all we know is the AlphaPeptDeep is used for predicting peptide properties. I assume you mean that AlphaPeptDeep in conjunction with a validation tool such as Percolator improves peptide identification performance? Please explain this explicitly in the manuscript.

pp 2, line 75 & 78 : “challenging samples like HLA peptides”, “HLA prediction model”

HLA needs to be defined before using the acronym. The second usage (HLA prediction) needs to be reworded. I assume you are predicting whether or not a given peptide will be presented, but as it is written it says the model is a “human leukocyte antigen prediction model”. The HLA is the protein.

pp 4, line 142 : “Prediction on GPU is still an order of magnitude faster.”

This is too exact of a statement. In general, inference on GPUs is going to be faster, often considerably faster, but how much faster depends on the hardware and the task at hand.

pp 4 lines 149 – 152 : What predefined models are available? Is the user able to change the architecture and hyperparameters of these models?

Figure 1

What is “Meta embedding”? This is mentioned in the online methods, but not anywhere near the figure.

pp 5, 6 lines 170-172 : “The MS2, RT, and CCS prediction models in AlphaPeptDeep are all publicly available in our Python modules (Fig. 2).”

Does this mean they are distributed as part of AlphaPeptDeep or does the user need to acquire them as part of a different Python module you built? Needs clarification.

pp 6, lines 174-175 : “making the prediction extremely fast”

How much faster relative to the Prosit-Transformer model?

pp 6, line 179 : “We trained and tested the MS2 models with tens of millions of spectra”

How many, exactly?

pp 6, lines 187-188 : “the human proteome took only 10 min on a regular GPU and 100 minutes on the CPU with multiprocessing”

This is really nice! Again, a comparison with existing tools would be helpful.

pp 7 lines 217-218 : “(timsTOF) in dda-PASEF mode, and achieved a PCC90 of 87.9%”

Was the PCC90 of 97% mentioned earlier for only Orbitrap data? You just say it is for ProteomeTools, which implies it also has timsTOF data in it. Do you have an explanation for why the performance against timsTOF spectra is 10% less? I am assuming the instrument type was included as a training feature.

pp 7, line 221 : “As expected, our pretrained model performed equally well across different organisms.”

Where are you showing this? Is it in fig 3a? It’s not very clear if the datasets in fig3a are from different species.

pp 7, lines 229-231 : “In a second training phase, we added a synthetic HLA dataset, which was also from ProteomeTools, into the training set and trained the model for additional 20 epochs (‘fine tuning the model’).”

Your exact procedure isn't clear to me from this sentence. Did you add the HLA data to the training set and then fine tune the model using the composite dataset? Or did you fine tune it using only the HLA dataset?

Is this "HLA-tuned" version of the MS2 prediction model included with AlphaPeptDeep? Or will the user need to do their own fine-tuning if they are doing immunopeptidomics experiments?

pp 8, line 275 : "drosophila data" – what drosophila data? This is the first time it is mentioned in the manuscript, so some more details are needed. And is the HeLa data the dataset you refer to at the beginning of the paragraph? The datasets should all be clearly defined early in the results.

pp 8, lines 276-277 : "For these peptides in the testing set, the R2 was 0.965 and 0.953"

I'm not really sure what you did here. Are "these" peptides the modified peptides? And is the testing set all of the search results from the HeLa and drosophila data? Also, my understanding from the preceding text is that the model was not trained with modifications in the data, so how did you deal with this in the test set? Were the peptides stripped of modifications before making predictions?

pp 8, lines 271-279 : What is the rationale behind evaluating the prediction performance of the RT and CSS models using two different methods? You trained both the RT and CSS models on data from HeLa samples, but then test the RT model on human and phosphopeptide datasets, and the CSS model on E. coli and drosophila datasets.

pp 8 : On page 8 you mention "regular" peptides a couple of times. How are you defining regular here? It's not clear from context, so it needs to be explicitly stated.

pp 9, line 293 : “we tested the pre-trained tryptic MS2 (phase 1 in Fig. 3a) and RT models using the 21 PTMs”

What are you referencing in Figure 3a? The data in this section is presented in Figure 4.

pp 9-10. lines 296-313 : This is impressive that you get such a significant improvement with such a small set of peptides for the transfer learning step. However, it is not very clear from the text what your experiment was in this section. The figure legend explains it okay, but the main text does seem to mention that you are comparing models transfer-trained on varying numbers of modified peptides, so it took me a while to catch on.

Figure 4c : This improvement is dramatic. It would be very interesting to see a random selection of peptides with other modifications. This could go in the extended figures.

pp 12, line 260 : “Such an approach has long been implemented in Percolator”

This is not how Percolator works. It does, as you say, help distinguish between correct and false identifications using semi-supervised learning, but there is no prediction of peptide properties happening in it. It uses an SVM, the input for which is a set of training features which incorporates most of the useful scoring information from a search engine (e.g. various search scores, mass error, charge state, etc.). Do you mean that Percolator has been used in conjunction with peptide prediction tools?

pp 12 lines : “However, due to the lack of support for arbitrary PTMs with DL models it has not been for open-search of HLA peptides.”

This would be the case for any open-search proteomics, right? Why the emphasis on HLA peptides?

pp 12, lines 363-365 : “Modern protein open-search engines like pFind can perform very fast unspecific peptide search without limiting the peptide mass window using the sequence tag technique, enabling the identification of unexpected PTMs.”

This sentence seems out of place. You have been talking about predicting MS2 spectra and RT for HLA peptides and using that to improve identifications in immunopeptidomics experiments. It seems strange to suddenly have a sentence describing pFind. If you are emphasizing the importance of looking for unexpected modifications in HLA data, and that AlphaPeptDeep is able to predict MS2 spectra and retention times for peptides with unexpected modifications, then you should reiterate this fact about AlphaPeptDeep, and then perhaps mention that pFind can be used in conjunction with it.

pp 12, line 367 : “AlphaPeptDeep fully supports the Percolator algorithm”

I understand that you are calculating some metrics characterizing the similarity between predicted and experimental MS2 spectra, retention times, and CCS for each PSM, and that these metrics are then added to the Percolator input file as additional features. The average reader might not have the computational background or the hands-on experience with Percolator to intuit this, so you should have a little explanation here. It doesn't have to be a lot, just a single sentence is probably enough to show everyone how the tools work together.

pp 12, lines 370-371 : “(~1 hour to rescore 16,812,335 PSMs from 424 MS runs using a PC with a GeForce RTX 3080 GPU”

How long does it take on a typical desktop? Many immunopeptidomics labs are not going to have a GPU workstation. I think even on a typical desktop it is probably still not going to be a bottleneck considering how long unspecific database searches of 424 MS runs would take.

pp 12, 382-389 : You should maybe emphasize here that all the search results in this paragraph are from MaxQuant. It sort of reads like AlphaPeptDeep is an alternative to MaxQuant, but really you are showing how much the MaxQuant results can be improved by putting them through AlphaPeptDeep (or Prosit).

pp 12, lines 391-398 : How automated is this process of taking results from open searches (or even closed searches) and rescoring them with AlphaPeptDeep? Does the user have to do a lot of formatting of peptide modifications in order to get it to work? Or does AlphaPeptDeep recognize files like pFind results, MaxQuant results, pepXML, idXML, or Percolator input files and automatically parse out the modification information for each PSM? If it is automated that would be great, because parsing and converting search results and especially modifications from one arbitrary format to another is often very painful.

pp 13, lines 409-410 : “We further used AlphaPeptDeep to inspect retention time and MS2 spectrum similarities (Extended Fig. 8)”

You need to say something about the conclusions you drew from this analysis, especially since the figure is not in the main text.

pp 13, lines 410-412 : “Note that the MS2 and RT models were only fine-tuned by at most 100 phospho-PSMs from eight RAW files (Online Methods), so most of the phosphopeptides from other RAW files were not used in fine-tuning.”

What other RAW files do you mean? I don't quite follow here.

pp 13, lines 412-414 : “Our method was also able to identify other PTMs associated with HLA peptides, such as cysteinylolation”

In the open-search pFind results when you applied AlphaPeptDeep, was the improvement for modified and unmodified peptides proportionally similar? Or did it result in a larger improvement for modified or for unmodified peptides?

p 15, lines 437-439 : “So far, these efforts have been restricted to experimental DIA libraries because analysis with a predicted HLA library is far more challenging than with an experimental one.”

I believe the Bassani-Sternberg group published some work comparing different strategies for immunopeptidomics DIA library generation, and one of their libraries was built with Prosit predictions (pmid: 33845167). However, I’m not sure if they used computational methods to come up with the peptide list for this library, or if the peptide list was based on experimental results.

p 15, lines 460-461 : “In brief, we first trained a pan-HLA prediction model with peptides from all known HLA types”

I assume this would be a binary prediction model: 1 for HLA-associated peptide, 0 for not. What did you use for peptide sequences which were labeled with “0”? Were they random? Biological in origin, but not appearing in any HLA peptide database? Or did the databases already include “not HLA” peptides?

p 15, line 461 : “all known HLA types”

There are approximately 25,000 named HLA alleles, but you later indicate that the model is trained on 94 HLA types (line 474). So it should just say it is trained on data from 94 HLA types.

p 15, lines 461-462 : “Normally, only a few HLA types are actually present in the samples from any given individual”

Up to 6 HLA alleles per individual

p 16, line 475 : “with 82% sensitivity.”

How did you calculate this sensitivity? Is it based on the direct-DIA results?

p 16, lines 484-486 : “The number of HLA peptides predicted by this model is 3M, which is comparable to the tryptic human proteome library.”

I would say that both 3M and 7M are comparable to 900,000 when compared against the original non-specific peptide space of 70M. Perhaps better to say that 3M is more comparable.

p 16, line 504-505 : “To simulate this situation, we removed all peptides of the dominant HLA-A*68:01 and used the rest to train a new pan-HLA model”

It is not explicitly clear here what from what dataset you are removing these peptides. I assume you mean the dataset that was used to train the pan-allele model?

How did you choose this particular HLA type to remove? It would seem best to remove an allele whose binding motif is distinct from all the other alleles in the training set, but this can be hard to do because there is a lot of overlap in the binding specificity of HLA alleles. Just looking through the first several alleles in the 94-allele dataset and comparing their binding motifs with A*68:01, I saw that A*11:01, A*11:02, A*33:01, A*33:03 and A*34:02 were similar at the anchor residues. There could be many more, so even though your allele is absent from the training set, the pan-HLA model might still recognize the peptides as HLA peptides. In general, this is a good thing, I think! But it means it might be difficult test the pipeline on truly “unseen” HLA alleles by removing a single allele from the dataset.

Possibly a better option would be test your pipeline on data from something non-human, like a mouse dataset. There are good predictors for mouse HLA presentation, so you should still be able to get a good idea about the sensitivity of the library you build. If you can find a DIA dataset for mouse, that is... I think maybe something came out of the Abersold group back in the early days of SWATH. There is the mouse atlas, but I don't recall if that has any DIA data in it or not.

Another possibility would be to go through all 94 alleles and look at their motifs (DTU Health Tech has a good motif viewer (look for the “Motif viewer” tab): <https://services.healthtech.dtu.dk/service.php?NetMHCpan-4.1>). You could remove every allele that resembles A*68:01. Or pick an allele with something really easy to spot, like the giant E at position 1 of B*40:01, and then remove everything that looks like that.

pp 21, lines 679-680 : “Model fine-tuning. 5,000 PSMs are randomly sampled from at most eight RAW files at 1% FDR to fine-tune the MS2, RT and CCS”

What metric is used for the FDR calculation? Does it depend upon FDR having been previously calculated by a tool, or can you specify a metric to use?

p 21, lines 688-689 : “61 score features”

In the supplemental data I see that there is only one feature for “Search engine score”. It is common for search engines to report multiple scores. Does AlphaPeptDeep choose a single score to include? Or do they all end up in the Percolator features?

Reviewer #3 (Remarks to the Author):

The manuscript first describes a deep learning framework that can learn and predict peptide properties - retention time, collisional cross sections and fragment ion intensities possibly for modified peptides. It also facilitates adaptation of a pre-trained general model to different tasks that involve the prediction of various peptide properties. The latter part of the manuscript describes how this framework can be used to set up a data analysis pipeline for HLA peptides and improve HLA peptide identification.

There is nothing new about AlphaPeptDeep itself. Almost all the main components have already been published: Deep learning has been applied and proven to be effective in predicting MS2

spectra and retention time only from peptide sequences; Transfer learning has been shown to be effective in predicting MS2 spectra of modified peptides, given a trained model for unmodified peptides; Rescoring peptide-spectrum matches using predicted MS2 spectra has been shown to improve peptide identification performance. While offering a user-friendly interface for transfer learning can be helpful, it is not worth another publication.

However, there are some intriguing aspects about its application to an HLA peptide analysis pipeline. AlphaPeptDeep was used to create a pan-HLA prediction model in the first place, and transfer learning was then adopted to classify and select HLA peptide sequences specific to particular samples, reducing the number of sequences required to generate a library of predicted spectra and eliminating the need for a DDA experiment to build a spectral library for DIA search.

While this approach to DIA-HLA analysis is interesting, the authors did not make it clear why the DIA experiment is superior to DDA in HLA peptide analyses. The primary goal of immunopeptidomics is the discovery of HLA peptide sequences. There are pros and cons for both the DDA and DIA experiments, and it has been and remains unclear which of the two gives better performance in peptide identification, if quantitation is not an issue. The DIA experiment is an overarching presumption in the proposed HLA peptide analysis pipeline, because DIA data analysis requires the construction of a spectral library, and its size needs to be controlled for various reasons (such the nonspecific digestion of HLA peptides, FDR control, etc.), necessitating the need for predicting HLA peptides that are known to be sample specific.

It is unclear how "DDA" was obtained when comparing the number of identified HLA peptides (Fig. 6b). In Fig. 5a, the authors demonstrated that the MaxQuant search with rescoring produced significantly improved outcomes. Therefore, "DDA" number in Fig. 6b would be a lot more informative if the rescoring was applied.

It is my opinion that the authors restructure the manuscript by focusing on the proposed HLA peptide analysis pipeline and its performance gain. It can certainly be added that AlphaPeptDeep, which offers a user-friendly interface to deep learning and transfer learning in the prediction of peptide properties, was very helpful in the process of building the pipeline.

Saturday, October 1, 2022

Point by point response to reviewers of “AlphaPeptDeep: A modular deep learning framework to predict peptide properties for proteomics”

Reviewer #1 (Remarks to the Author):

The manuscript presents a modular deep learning framework to predict the LC-MS-related properties of a given peptide sequence including retention time, ion mobility, and fragment pattern. This framework does not only provide users with high-performance pre-trained models, but also the capability of deploying new neural network architectures for prediction tasks. It may generate project-specific models via transfer learning, which improves the performance greatly in a variety of applications such as HLA peptide identification from DDA and DIA data. Overall, I think it is a valuable framework for exploring deep learning solutions in MS-based proteomics.

Reply: We thank the reviewer for the positive comments about our AlphaPeptDeep framework.

What I concern is about the validity of those additionally identified HLP peptides. As the ground truth is lacking, it is hard to know they are true or false positives. Hence, it might be a good idea to have several case studies presented here, for instance randomly select a small number of HLP peptides, mirror-plot their experimental and predicted spectra, and/or draw their precursor and fragment ion elution profiles in case of DIA data. Such manual inspection shall reveal how reliable those identifications are to some extent.

Reply: We do agree that manually validation is very important for HLA peptides as well as for proteomics studies in general. In the revision we have followed the reviewer’s suggestion for visualizing our predictions in comparison to the experimental results. To this end, for both DDA and DIA data, we built on a project for RAW data access, called AlphaRaw (<https://github.com/MannLabs/alpharaw>). Note that AlphaRaw is not public yet but can already be installed via ‘pip install alpharaw’ (this is because we are still updating this package frequently and are doing more testing). Based on AlphaRaw and AlphaPeptDeep, we added to our visualization tool alphaviz (https://github.com/MannLabs/alphaviz/tree/peptdeep_latest), so that it can now visualize MS2 mirror plots and elution profile plots of Thermo DDA and DIA results. For a tutorial and examples please see https://github.com/MannLabs/alphaviz/blob/peptdeep_latest/nbs/tutorial_DIANN_alphabase.ipynb. This module enabled us to validate the DDA and DIA results in this paper, and it can also be used with other search engines.

To validate HLA DDA data with this tool, we randomly selected 300 HLA peptides including 44 phosphorylated ones from the A1102, B4201, B5501 and C1202 HLA allele types in Fig. 5c. Results showed a PCC90 of 90% (meaning 90% of correlation values are greater than 90%) for unmodified

and phosphorylated HLA peptides. A typical MS2 mirror plot is shown below ('-mod' in the annotation text refers to -98 Da phospho-neutral loss). However, we have put all these 300 plots in Suppl. Data. 4, and updated the text in DDA rescoring section of the revised manuscript:

“We further used AlphaPeptDeep to inspect retention time and MS2 spectrum similarities. The results showed 80% PCC90 of phospho-HLA PSMs which were close to unmodified ones, and RT differences close to zero (Extended Fig. 8). **Furthermore, we manually validated 300 randomly selected HLA peptides including 44 phospho HLA peptides using an extension to AlphaViz. Their annotated and mirrored MS2 plots can be found in Suppl. Data 4. This independently verified our model and assignments.** Note that the MS2 and RT models were only...

To validate HLA DIA data, we plotted both of the elution profiles and MS2 mirror annotation for 100 randomly selected HLA peptides. In the DIA data there was more interference by peptides within the same DIA window, but we still observed good PCC values for 60% of the HLA peptides (PCC80 = 60%). The elution and MS2 mirror plots are in Suppl. Data 5, and we updated the text in the revised manuscript as follows:

“...indicating that our additionally identified HLA peptides were reliable at the same level. **We also manually checked 100 randomly selected peptides by plotting their elution profiles and mirrored MS2 annotations (Suppl. Data 5). ...”.**

The figure below shows a peptide with only 0.8 PCC but very consistent fragment elution profiles.

_PEDSKAVLL_2 PCC=0.797

Reviewer #2 (Remarks to the Author):

The authors present a Python package, AlphaPeptDeep. The package is meant to make the use of deep learning methods in MS-based proteomics more accessible to non-experts. The package also comes equipped with a set of pre-trained models for MS2 fragmentation, retention time, and CSS prediction, all of which can be fine-tuned for a user's specific dataset. The package is evaluated in several experiments, with special attention given to immunopeptidomics MS experiments.

I am impressed with the idea behind AlphaPeptDeep and the authors' work in putting it together. However, I think that there are issues to be addressed in both the manuscript and the online documentation for the tool before it can be considered for publication.

Reply: We thank the reviewer the positive comments and the thorough examination of our paper, which has made it much stronger in our opinion. In particular, we agree that the online documentation should be improved. During the revision, we worked a lot more on these documentation parts as discussed in the following replies.

In general, I think there needs to be more in the manuscript on how AlphaPeptDeep is used. A lot of the manuscript is evaluating trained models, but the actual thing that is new and novel here is the Python package that was used to create all those models. That is the exciting thing, that someone who knows some Python can install a single package and then build deep learning models for predicting pretty much any peptide property they can come up with so long as it is dependent upon peptide sequence. Obviously, we need to know that models created for standard tasks like MS2 prediction work as expected, but more emphasis should be given to the novel aspect here. The models themselves are not the novelty in this work (with the exception of the HLA library generation, which is quite neat and does showcase how new things can be done with AlphaPeptDeep).

Reply: Thank you for the comments. It is true that user specification is important. We hope the reviewer also agree that - as we are now working on open-source projects - it is better to place the exact technical specifications on GitHub rather than the manuscript. Because the package is still growing, this insures that the package will still work as expected in the future. Therefore, as the reviewer suggested, we have now improved our documentation (using github pages and jupyter notebooks).

We also agree that MS2/RT/CCS prediction is not novel as such. That said, AlphaPeptDeep does have novel features, such as supporting all PTM types in different models. To show the power of this modular framework, we evaluated it not only for the standard prediction problems (MS2/RT/CCS) but also for novel ones (HLA prediction in proteomics). We believe that integrating state of the art MS2/RT/CCS models with different architectures makes the framework very versatile and the manuscript more completed.

I have worked a lot with different machine learning-based tools for proteomics, and user-friendliness has almost always been an issue. AlphaPeptDeep intends to provide an easy-to-use package for developing, training, fine-tuning, and running predictors for any peptide property which depends on peptide sequence. I think AlphaPeptDeep is on the right track to do this. The authors want to make an easy way to train arbitrary deep learning models for tasks in proteomics, which is a fantastic idea. They have really hit on the fact that predicting peptide properties is just a sequence-to-sequence or sequence-to-scalar task, and that many problems can be addressed with models having the same basic structure. However, I think there needs to be much more in the way of documentation and online tutorials for it to really be user-friendly and approachable for non-experts. This is an integral part of making useful software tools.

Reply: We completely agree that there are a lot of tools that are powerful in principle but rarely used because they are not user friendly. With AlphaPeptDeep and the other parts of the AlphaPept

ecosystem, it is our explicit goal to avoid this situation. Therefore, we appreciate the reviewers detailed suggestions and have tried our best to follow them as much as we can since they are reasonable and useful to improve our work.

I have a lot of comments and questions on the manuscript, which will follow. I hope they are not taken harshly, because I think the tool has a lot of potential. However, I think work needs to be done to clarify some parts of the manuscript and the use of the tool. First, however, I will discuss my experience using AlphaPeptDeep, which I think illustrates the need for more comprehensive documentation.

Please note that my comments below reflect the online documentation at 19:00 hr CET August 13, 2022. If changes were being made to the documentation on an ongoing basis, I might have missed those changes.

Reply: We improved the documentation with more than 60 GitHub commits in September. Please check our GitHub pages for the latest documentation (<https://mannlabs.github.io/alphapeptdeep>). In summary, we built documentation for important functions and classes in AlphaPeptDeep. For example, in https://mannlabs.github.io/alphapeptdeep/pretrained_models.html, we now have the input and output details of APIs for users of our ModelManager class, as shown below:

ModelManager.predict_ms2 source

```
ModelManager.predict_ms2 (precursor_df:pandas.core.frame.DataFrame,  
                           batch_size:int=512, reference_frag_df:pandas.co  
                           re.frame.DataFrame=None)
```

Predict MS2 for the given precursor_df

	Type	Default	Details
precursor_df	DataFrame		Precursor dataframe for MS2 prediction
batch_size	int	512	Batch size for prediction. Defaults to mgr_settings['predict']['batch_size_ms2']
reference_frag_df	DataFrame	None	If precursor_df has 'frag_start_idx' pointing to reference_frag_df. Defaults to None
Returns	DataFrame		Predicted fragment intensity dataframe. If there are no such two columns in precursor_df, it will insert 'frag_start_idx' and frag_end_idx in precursor_df pointing to this predicted fragment dataframe.

We also provided several notebooks on GitHub including:

1. Library prediction from fasta files:
https://github.com/MannLabs/alphapeptdeep/blob/main/nbs/tutorial_speclib_from_fasta.ipynb

2. Library translation from AlphaPeptDeep to DiaNN/Spectronaut:
https://github.com/MannLabs/alphapeptdeep/blob/main/nbs/alphapeptdeep_hdf_to_tsv.ipynb
3. Training of the HLA prediction models:
https://github.com/MannLabs/alphapeptdeep/blob/main/nbs/HLA_peptide_prediction.ipynb
4. A Percolator example:
https://github.com/MannLabs/alphapeptdeep/blob/main/nbs_tests/test_percolator.ipynb

We hope this will help researchers to use AlphaPeptDeep in their own projects.

The manuscript points the reader to a GitHub page for AlphaPeptDeep. I followed this and found a nicely organized repository with some well-written instructions for installing the tool from the command line. I also saw that there is a GUI available which could be downloaded from the releases section. I am a bioinformatician and programmer, but being a human I am also somewhat lazy and so I thought the GUI would be the simplest thing to try out first. After downloading it I tried to build a predicted spectral library. I typically work with peptides and not whole proteins, so I tried using the peptide table input, but ran into the problem that there was no description of what the peptide table should look like. I looked on the GitHub repository, but again there was no explanation and no examples. There is a section for modification information in the GUI, but it is more like what you expect in a search engine: a list of fixed and variable modifications. Since there is no description or examples of how to format a peptide table, I thought that perhaps it just takes a peptide list and then iterates through all the possible charge states and variable modifications. However, when I tried loading the peptide list, the GUI printed out an error message that it could not determine the file delimiter (though it is very nice that the GUI shows the error traceback). Obviously, it is expecting a delimited file (like as CSV or TSV), but I don't know what to give it and I can't find out from the available resources.

Since the peptide table input could not be used, I tried giving it a FASTA file with a few proteins in it. That worked, and very well! Within minutes I had a predicted spectral library, and it looked exactly how it should. That satisfied me that the package generally works as described, and that the problem is that there was simply no way to know what kind of input the tool was expecting for the peptide table (except for the FASTA).

Reply: Thank you for pointing this out and please see the answers just above, too. In the revision, we have also improved the GUI tool. We believe that the GUI is easy to use, so we did not discuss much about it, see <https://mannlabs.github.io/alphapeptdeep/#gui>. For the peptide_table format, we now displayed it on https://mannlabs.github.io/alphapeptdeep/#sequence_table. The GUI will also notify users to check this link for the file format when they select 'sequence_table' or 'peptide_table'.

"Cannot determine CSV delimiter" is indeed a bug, we have fixed it.

Despite the fact that the GUI is prominently mentioned on the GitHub page, it was not mentioned in the manuscript. Thus, I assumed that the GUI is in development and that is why there are no built-in instructions and no online documentation. I next tried the CLI. It was easily installed into a clean Python virtual environment using the instructions in the GitHub repository (`pip install peptdeep`).

Reply: Indeed, the GUI was still under development at the time of submission. Even now, we are still improving on the GUI for rescoring as we are expanding to different search engines and different RAW data from different vendors. Rescoring via CLI with AlphaPept/MaxQuant/pFind and Thermo RAW data now works well.

When I tried running the CLI, it started fine and showed the expected help screen:

```
...
```

```
$ peptdeep -h
Usage: peptdeep [OPTIONS] COMMAND [ARGS]...
```

Options:

```
-v, --version Show the version and exit.
-h, --help Show this message and exit.
```

Commands:

```
gui Start graphical user interface.
install-models Install peptdeep pretrained models.
library Predict library for DIA search.
rescore Rescore DDA results.
transfer Transfer learning for different data types.
```

```
...
```

I next wanted to know how to generate a library, so I tried getting help on that function, but the CLI told me nothing helpful:

```
...
```

```
$ peptdeep library -h
Usage: peptdeep library [OPTIONS] SETTINGS_YAML
```

Predict library for DIA search.

Options:

```
-h, --help Show this message and exit.
```

```
...
```

It seems to expect a YAML file with some settings in it (as well as some options), but there are no example YAML files that I could easily find on the GitHub page, nor any documentation on what it should look like.

The same commands for “rescore” and “transfer” gave similar output. The only documentation in the GitHub repository for the CLI is this: “It is possible to get help about each function and their

(required) parameters by using the -h flag.” There is no other documentation, tutorials, or examples.

Reply: We are sorry that we did not provide documentation for this yaml file. We now provide a command “peptdeep export-settings /path/to/settings.yaml” to generate a template yaml file to edit. Users can also export it from GUI’s “settings” panel. And the structure of the yaml file is also discussed on <https://mannlabs.github.io/alphapeptdeep/#cli>. We can also edit the yaml file and then upload it in GUI or run the task via CLI for different tasks. Details can be found on <https://mannlabs.github.io/alphapeptdeep/#export-settings>. CLI docs for different tasks can be found on:

Library prediction: <https://mannlabs.github.io/alphapeptdeep/#library>

Transfer learning: <https://mannlabs.github.io/alphapeptdeep/#transfer>

DDA rescoring: <https://mannlabs.github.io/alphapeptdeep/#rescore>

If one is technically inclined, one could load AlphaPeptDeep in a running Python environment and use it from there. There is documentation for the API linked from the GitHub repository, but it is also lacking in examples or tutorials. The API documentation itself is pretty sparse, as there are no definitions given for any function parameters. Thus, if one wants to really know how to use the tool, they will need to go to the source code. I think very few people are going to go that far…

Reply: In the new version, we used nbdev2, a tool for ‘literate programming’ and building documentation from jupyter notebooks. For an example of this, please see the output for the `ModelManager.predict_ms2` above in this reply.

It seems, at least based on the current state of the documentation, that the tool is unusable by people who do not already know how to use it. The idea behind AlphaPeptDeep is great. It will be wonderful for people who have a little coding experience, but not enough to be comfortable diving into PyTorch or Tensorflow, to be able to develop and train a deep learning model for prediction of arbitrary peptide properties. But without proper documentation and tutorials to tell them how to do it, there is no way for them to get started. In short, even as an expert I would not be able to easily use AlphaPeptDeep to reproduce the DL models produced by the authors. As that production is an important part of the results, I believe this issue needs to be addressed prior to any possible publication.

Having said all that, I have seen enough that I believe the tool does work, and that the authors have put a lot of time and effort into it. I hope they will put together some excellent documentation to show others how to use AlphaPeptDeep, because it does seem like something that could really benefit the proteomics community.

Reply: We really thank the reviewer for checking not only the manuscript but also the tool itself. All the comments are very useful and enabled us to improve alphapeptdeep, especially in the documentation aspect.

As already mentioned by example above, we have worked on documentation during these months

to cover as many AlphaPeptDeep features as possible. Now we think both the documentation and tutorials are up to a high standard. In summary:

- The usage specification of GUI and CLI are easier to access, see our readme on GitHub: <https://github.com/MannLabs/alphapeptdeep> as well as our GitHub pages (<https://mannlabs.github.io/alphapeptdeep>).
- For developers, the API documentation is also improved, see <https://mannlabs.github.io/alphapeptdeep>.
- We also now provide several example notebooks on <https://github.com/MannLabs/alphapeptdeep/tree/main/nbs> including:
 1. library prediction from fasta:
https://github.com/MannLabs/alphapeptdeep/blob/main/nbs/tutorial_speclib_from_fasta.ipynb
 2. library translation from AlphaPeptDeep to DiaNN/Spectronaut:
https://github.com/MannLabs/alphapeptdeep/blob/main/nbs/alphapeptdeep_hdf_to_tsv.ipynb
 3. training of the HLA prediction models:
https://github.com/MannLabs/alphapeptdeep/blob/main/nbs/HLA_peptide_prediction.ipynb
 4. percolator example:
https://github.com/MannLabs/alphapeptdeep/blob/main/nbs_tests/test_percolator.ipynb

The remainder of my comments will be on the manuscript itself.

pp 2, lines 51-52 : “This is also the case for predicting the fragment intensities in the MS2 spectra, where DL models such as our previous model pDeep, DeepMass:Prism, Prosit and many subsequent ones now represent the state-of-the-art.”

This sentence makes it sound like all three of these tools might come from the same group, which they do not.

Reply: This was not our intention. We changed this to “... DeepMass:Prism, Prosit, and our previous model pDeep ...”

pp 2, lines 75-76 : “AlphaPeptDeep dramatically boosts performance of peptide identification for data-dependent acquisition”

This implies that AlphaPeptDeep is directly involved in the identification process. Most readers will probably understand what you are getting at, but at this point all we know is the AlphaPeptDeep is used for predicting peptide properties. I assume you mean that AlphaPeptDeep in conjunction

with a validation tool such as Percolator improves peptide identification performance? Please explain this explicitly in the manuscript.

Reply: Yes, indeed. We have changed the text to “... AlphaPeptDeep coupled with its built-in Percolator implementation dramatically ...”.

pp 2, line 75 & 78 : “challenging samples like HLA peptides”, “HLA prediction model”

HLA needs to be defined before using the acronym. The second usage (HLA prediction) needs to be reworded. I assume you are predicting whether or not a given peptide will be presented, but as it is written it says the model is a “human leukocyte antigen prediction model”. The HLA is the protein.

Reply: This is a good point. We modified this sentence to “... such as a model for human leukocyte antigen (HLA) peptide prediction ...”

pp 4, line 142 : “Prediction on GPU is still an order of magnitude faster.”

This is too exact of a statement. In general, inference on GPUs is going to be faster, often considerably faster, but how much faster depends on the hardware and the task at hand.

Reply: Changed to “On our data sets and hardware, prediction on GPU was about an order of magnitude faster.”

pp 4 lines 149 – 152 : What predefined models are available? Is the user able to change the architecture and hyperparameters of these models?

Reply: Yes, this is what we aim for. Currently, users can either choose transformer or the LSTM architecture from the model shop. Users can change the number of NN layers, the hidden layer size and dropout values while training. We now changed the text to:

“AlphaPeptDeep provides several model templates based on transformers and LSTM architectures in the “model shop” module to develop new DL models and also allows choosing hyperparameters from scratch for classification or regression with very little code.”

Figure 1

What is “Meta embedding”? This is mentioned in the online methods, but not anywhere near the figure.

Reply: Explanation added into the figure caption:

“Meta embedding refers to the embedding of meta information such as precursor charge states,

collisional energies, instrument types, and other non-sequential inputs.”

pp 5, 6 lines 170-172 : “The MS2, RT, and CCS prediction models in AlphaPeptDeep are all publicly available in our Python modules (Fig. 2).”

Does this mean they are distributed as part of AlphaPeptDeep or does the user need to acquire them as part of a different Python module you built? Needs clarification.

Reply: Yes, it is the latter. These models are not in the AlphaPeptDeep python package, but they will be automatically downloaded from <https://github.com/MannLabs/alphapeptdeep/releases/tag/pre-trained-models> when using AlphaPeptDeep for the first time. We have changed the statement accordingly:

“The MS2, RT, and CCS prediction models (Fig. 2) are released on our GitHub repository and will be automatically imported into AlphaPeptDeep when using the package for the first time.”

pp 6, lines 174-175 : “making the prediction extremely fast”

How much faster relative to the Prosit-Transformer model?

Reply: On the same test sets, MS2 prediction is 40 times faster than Prosit-Transformer using the same GPU workstation. We have added related plots for testing 1.4M peptides in Extended Fig. 3a and mentioned this in the revised text.

“Testing by the same 1.4M peptides on the same GPU workstation showed that fragment intensity prediction of AlphaPeptDeep is 40 times faster than Prosit-Transformer (35 seconds vs 24 minutes, Extended Fig. 3).”

pp 6, line 179 : “We trained and tested the MS2 models with tens of millions of spectra”

How many, exactly?

Reply: # training = 23,846,809, # testing = 20,435,493. See Suppl. Data 1. We have updated the number with ~40 million in the main text:

“We trained and tested the MS2 models with ~40 million spectra from a variety of instruments, collision energies and peptides”

pp 6, lines 187-188 : “the human proteome took only 10 min on a regular GPU and 100 minutes on the CPU with multiprocessing”

This is really nice! Again, a comparison with existing tools would be helpful.

Reply: We have compared the MS2 prediction speed with Prosit-Transformer in Extended Fig. 3a in the reply just above. We did not compare the speed of library generation as:

1. Prosit-Transformer does not support fasta files; and
2. Prosit-Transformer does not include fragment m/z calculation and RT/CCS prediction to generate the libraries

pp 7 lines 217-218 : “(timsTOF) in dda-PASEF mode, and achieved a PCC90 of 87.9%”

Was the PCC90 of 97% mentioned earlier for only Orbitrap data? You just say it is for ProteomeTools, which implies it also has timsTOF data in it. Do you have an explanation for why the performance against timsTOF spectra is 10% less? I am assuming the instrument type was included as a training feature.

Reply: Yes, this is indeed a little bit confusing. The original ProteomeTools data were measured by the Kuster lab on a Lumos instrument. We then obtained the same sample from Kuster lab and measured it on the timsTOF instrument. We have updated the texts accordingly to make this clearer:

“Overall, on ProteomeTools data measured with different collisional energies on the Lumos mass spectrometer, 97% of all significantly matching PSMs had Pearson correlation coefficients (PCC) of the predicted vs. the measured fragment intensities of at least 90%.”

And in the caption of Fig. 3a, we also added the text:

“PT1, PT2, and PT-HLA were all measured with Lumos by the Kuster lab.”

pp 7, line 221 : “As expected, our pretrained model performed equally well across different organisms.”

Where are you showing this? Is it in fig 3a? It's not very clear if the datasets in fig3a are from different species.

Reply: To make it clear, we have updated the references:

“As expected, our pretrained model performed equally well across different organisms, as demonstrated by PXD019086-Drosophila and -Ecoli in Fig. 3a. Interestingly, it did almost as well on

chymotrypsin or GluC-digested peptides although it had not been trained on them (PXD004452-Chymo and -GluC in Fig. 3a). ”

pp 7, lines 229-231 : “In a second training phase, we added a synthetic HLA dataset, which was also from ProteomeTools, into the training set and trained the model for additional 20 epochs (‘fine tuning the model’).”

Your exact procedure isn’t clear to me from this sentence. Did you add the HLA data to the training set and then fine tune the model using the composite dataset? Or did you fine tune it using only the HLA dataset?

Is this “HLA-tuned” version of the MS2 prediction model is included with AlphaPeptDeep? Or will the user need to do their own fine-tuning if they are doing immunopeptidomics experiments?

Reply: We used transfer learning based on the HLA training datasets of phase 1. To make the statement more accurate, we now changed to “append”: “we appended a synthetic HLA dataset ...”

Yes, the default pre-trained model in AlphaPeptDeep is from phase 3. We also added the text: “The final model was saved as the default pre-trained model (“generic”) in the AlphaPeptDeep package.” before the subsection “Prediction performance of the AlphaPeptDeep models for RT and CCS”.

pp 8, line 275 : “drosophila data” – what drosophila data? This is the first time it is mentioned in the manuscript, so some more details are needed. And is the HeLa data the dataset you refer to at the beginning of the paragraph? The datasets should all be clearly defined early in the results.

Reply: Thank you for pointing this out. To address these issues, we decided to add a new table in the manuscript to show the dataset information for RT/CCS model training and testing. We hope this makes it easier for readers.

Dataset	Search	Modifications	Usage	Description
RT model				
HeLa	MaxQuant	regular	training	Trypsin and LysC HeLa peptides. Ref ²⁶
PHL		regular	testing	Pan human library. Ref ³⁵
Phos-U2OS	Spectronaut	regular and phos	testing	Phosphopeptides of U2OS. Ref ³⁶
CCS model				
HeLa	MaxQuant	regular	training	Same as HeLa in RT section
E. coli	MaxQuant	regular	testing	E. coli peptides. Ref ²⁶
Yeast	MaxQuant	regular	testing	Yeast peptides. Ref ²⁶
HeLa-Open	Open-pFind	all possible PTMs	testing	Same as HeLa in RT section. Only peptides with nonregular modifications were kept after open-search for testing
Drosophila-	Open-pFind	all possible PTMs	testing	Drosophila peptides. Ref ²⁶ . Only peptides

Open				with nonregular modifications were kept after open-search for testing
------	--	--	--	---

Table 1. Dataset information used to train and test RT/CCS models. ‘regular’ in the ‘Modifications’ column refers to unmodified, Oxidation@M, Carbamidomethyl@C and Acetyl@Protein N-term. The ‘Search’ column with ‘Open-pFind’ means that we re-analyzed the MS data with Open-pFind³⁴ (Online Methods), and only peptides with nonregular modifications were kept for testing. Otherwise, the search results were downloaded from the original publications of the datasets.

pp 8, lines 276-277 : “For these peptides in the testing set, the R2 was 0.965 and 0.953”

I’m not really sure what you did here. Are “these” peptides the modified peptides? And is the testing set all of the search results from the HeLa and drosophila data? Also, my understanding from the preceding text is that the model was not trained with modifications in the data, so how did you deal with this in the test set? Were the peptides stripped of modifications before making predictions?

Reply: See just above. We added Table 1 to help understand the datasets.

For PTMs unseen by the model, we do not need to do any stripping. As we used chemical composition-based embedding for PTMs, any PTM will be embedded by its chemical compositions. And “the model was not trained with modifications” means that the model was only trained with “regular” (Oxidation@M and Carbamidomethyl@C) PTMs. But we can still use the model to predict “nonregular” ones.

pp 8, lines 271-279 : What is the rationale behind evaluating the prediction performance of the RT and CCS models using two different methods? You trained both the RT and CSS models on data from HeLa samples, but then test the RT model on human and phosphopeptide datasets, and the CSS model on E. coli and drosophila datasets.

Reply: Similar to the previous question, we trained the model with regular PTMs, and we would like to see how good the model and transfer learning will be for other LC conditions and PTMs like phosphorylation.

So we trained the RT model on HeLa and tested it on:

1. Pan human library (regular). From human sample as well but with different sources.
2. Phosphorylation. Nonregular PTMs.

We also applied a similar idea for CCS model testing. We trained the model on HeLa and tested on:

1. Regular peptides from E. coli and yeast. Different species from the training data.
2. Peptides with nonregular PTMs from HeLa and drosophila. We did not use E. coli and yeast for open-search because E. coli and yeast are thought to have fewer complex modifications.

To make this clear in the revised manuscript, we updated the CCS evaluation paragraph as:

“While the CCS model was trained on regular human peptides from the same HeLa dataset as RT model training, we tested the trained model on different types of data (Table 1). First, we tested on regular peptides but from non-human species. Here we used E. coli and yeast peptides from the same instrument in the same publication. For these regular peptides the CCS model achieved an $R^2 > 0.98$ of the predicted and detected CCS values. Second, we tested on peptides with different modifications. For modified peptides from HeLa-Open and Drosophila-Open datasets (Table 1), the R^2 was 0.965 and 0.953, respectively, a prediction accuracy quite close to the one for regular peptides, even for unexpected modifications. The predicted CCS values can be converted to ion mobilities on the Bruker timsTOF using the Mason Schamp equation.”

pp 8 : On page 8 you mention “regular” peptides a couple of times. How are you defining regular here? It's not clear from context, so it needs to be explicitly stated.

Reply: We added the following explanation to the revised text “ ‘Regular peptides’ refers to unmodified peptides or modified peptides only containing Oxidation@M, Carbamidomethyl@C and Acetyl@Protein N-term”.

pp 9, line 293 : “we tested the pre-trained tryptic MS2 (phase 1 in Fig. 3a) and RT models using the 21 PTMs”

What are you referencing in Figure 3a? The data in this section is presented in Figure 4.

Reply: We referred to the model from training phase 1 in Fig. 3a. We need a pre-trained MS2 model to test PT-21PTM. The model from phase 3 in Fig. 3a is not appropriate for testing because it already incorporates phosphorylation and diGly which are also included in PT-21PTM dataset. Both models from phase 1 and 2 are OK to use here, we just selected the phase 1 model. To clarify this we updated the text:

“To further demonstrate the powerful and flexible support for PTMs in AlphaPeptDeep, we tested the **pre-trained tryptic MS2 model (model of phase 1 in Fig. 3a) and RT model** using the 21 PTMs, which were synthesized based on 200 template peptide sequences”

pp 9-10. lines 296-313 : This is impressive that you get such a significant improvement with such a small set of peptides for the transfer learning step. However, it is not very clear from the text what your experiment was in this section. The figure legend explains it okay, but the main text does seem to mention that you are comparing models transfer-trained on varying numbers of modified peptides, so it took me a while to catch on.

Reply: Thank you! In the revised manuscript, we added “We applied transfer learning for each PTM type using 10 or 50 training peptides with different charge states and collision energies, reserving the remaining ones with the same PTM for testing of our transfer learned models. Furthermore, we also trained with 80% of the peptides and tested on the remaining 20% (Fig. 4a).”

Figure 4c : This improvement is dramatic. It would be very interesting to see a random selection of peptides with other modifications. This could go in the extended figures.

Reply: We agree and added Suppl. Data 3 and updated the text in the revised manuscript:
“We also generated mirrored MS2 plots for ten randomly selected peptides of each PTM type before and after transfer learning, see Suppl. Data 3.”

One example is shown below. This is also interesting that, as there is an Arg at position 2, so the pre-trained model predicted that b ions will have high intensities. However, deamidation on Arg will change the fragmentation pattern of Arg itself, making b ions lower than expected. This can be learned by transfer learning. There are many such examples in Suppl. Data 3.

FR(+1)GFM(+16)QK,2 PCC=0.003

FR(+1)GFM(+16)QK,2 PCC=0.997

pp 12, line 260 : “Such an approach has long been implemented in Percolator”

This is not how Percolator works. It does, as you say, help distinguish between correct and false identifications using semi-supervised learning, but there is no prediction of peptide properties happening in it. It uses an SVM, the input for which is a set of training features which incorporates most of the useful scoring information from a search engine (e.g. various search scores, mass error, charge state, etc.). Do you mean that Percolator has been used in conjunction with peptide prediction tools?

Reply: We modified the text to “Such an approach has been implemented in tools coupled with Percolator to re-score PSMs, which increases the sensitivity at the same FDR level.”

pp 12 lines : “However, due to the lack of support for arbitrary PTMs with DL models it has not been for open-search of HLA peptides.”

This would be the case for any open-search proteomics, right? Why the emphasis on HLA peptides?

Reply: We tried not to overclaim. However, as we have not been able to find any DL+Percolator method for open-search, we now removed “HLA peptides”.

“But due to the lack of support for arbitrary PTMs with DL models this has not been implemented for open-search. However, AlphaPeptDeep is now able to predict the properties of arbitrarily modified peptides, and even HLA peptides with unexpected PTMs.

pp 12, lines 363-365 : “Modern protein open-search engines like pFind can perform very fast unspecific peptide search without limiting the peptide mass window using the sequence tag technique, enabling the identification of unexpected PTMs.”

This sentence seems out of place. You have been talking about predicting MS2 spectra and RT for HLA peptides and using that to improve identifications in immunopeptidomics experiments. It seems strange to suddenly have a sentence describing pFnd. If you are emphasizing the importance of looking for unexpected modifications in HLA data, and that AlphaPeptDeep is able to predict MS2 spectra and retention times for peptides with unexpected modifications, then you should reiterate this fact about AlphaPeptDeep, and then perhaps mention that pFind can be used in conjunction with it.

Reply: Thank you for pointing this out. We have changed the statement accordingly:

“This feature is intended to boost the identification of HLA peptides in conjunction with modern open-search engines like pFind, which identifies unexpected PTMs by using the sequence tag technique.”

pp 12, line 367 : “AlphaPeptDeep fully supports the Percolator algorithm”

I understand that you are calculating some metrics characterizing the similarity between predicted and experimental MS2 spectra, retention times, and CCS for each PSM, and that these metrics are then added to the Percolator input file as additional features. The average reader might not have the computational background or the hands-on experience with Percolator to intuit this, so you should have a little explanation here. It doesn't have to be a lot, just a single sentence is probably enough to show everyone how the tools work together.

Reply: As mentioned in Online Methods (rescore section), we re-implemented the Percolator algorithm in AlphaPeptDeep based on the powerful machine learning ecosystem of python (e.g. scikit-learn). But the percolator idea came from Lucas Kall, et al in 2007, and we did not want to overclaim, which is why we said “supports” here. Now we changed the text to:

“AlphaPeptDeep applies the semi-supervised Percolator algorithm on the output of search engines, rescoring PSMs to better discriminate true identifications from false ones based on deep learning predicted parameters (Online Methods). Rescoring for the open-search is also supported.”

pp 12, lines 370-371 : “(~1 hour to rescore 16,812,335 PSMs from 424 MS runs using a PC with a GeForce RTX 3080 GPU”

How long does it take on a typical desktop? Many immunopeptidomics labs are not going to have a GPU workstation. I think even on a typical desktop it is probably still not going to be a bottleneck considering how long unspecific database searches of 424 MS runs would take.

Reply: Interesting question. The running time without GPU was ~3.5 hours for the same 16,812,335

PSMs, while Open-pFind took more than a week to search the RAW data. We now updated this in the revised manuscript:

“Running without the GPU on the same PC took ~3.5 hours, whereas non-specific open-search for this many spectra took more than a week, meaning that the rescoring by AlphaPeptDeep is not a bottleneck for HLA peptide search.”

pp 12, 382-389 : You should maybe emphasize here that all the search results in this paragraph are from MaxQuant. It sort of reads like AlphaPeptDeep is an alternative to MaxQuant, but really you are showing how much the MaxQuant results can be improved by putting them through AlphaPeptDeep (or Prosit).

Reply: We have added a sentence to make this clear: “The MaxQuant PSMs and Prosit-rescored PSMs were downloaded from ref 33 and the MaxQuant PSMs were rescored by AlphaPeptDeep for comparison.”

pp 12, lines 391-398 : How automated is this process of taking results from open searches (or even closed searches) and rescoring them with AlphaPeptDeep? Does the user have to do a lot of formatting of peptide modifications in order to get it to work? Or does AlphaPeptDeep recognize files like pFind results, MaxQuant results, pepXML, idXML, or Percolator input files and automatically parse out the modification information for each PSM? If it is automated that would be great, because parsing and converting search results and especially modifications from one arbitrary format to another is often very painful.

Reply: For most of the analysis, this process is automated as modification name translation is automatically done by AlphaPeptDeep. This is due to our module AlphaBase, which uses UniMod names for all modification information, and designates the modified amino acids by “@”, e.g Oxidation@M, Acetyl@Protein N-term...

For different search engines (MaxQuant, DiaNN, AlphaPept, pFind, Spectronaut), we have a pre-defined modification map to translate their regular modifications (Phospho, Oxidation, Carbamidomethyl, Acetyl, GlyGly) into AlphaBase format. We also allow developers to define their own modification map for different readers. See psm_reader in AlphaBase repo: https://github.com/MannLabs/alphabase/tree/main/alphabase/psm_reader. PepXML is not fully supported, but we support MSFragger’s PepXML, see https://github.com/MannLabs/alphabase/blob/main/alphabase/psm_reader/msfragger_reader.py#L80.

We now alert the user to this feature in the revised manuscript:

AlphaBase uses UniMod modification names to represent modifications, and designates the modified amino acids by “@”, e.g Oxidation@M, Acetyl@Protein N-term, etc. It provides result readers to import DDA and DIA search engines (e.g. AlphaPept, MaxQuant, DIA-NN, Spectronaut), and translate common modification names (e.g. oxidation, acetylation, carbamidomethylation, phosphorylation, di-gly) into AlphaBase format. Users can also provide modification dictionaries to tell AlphaBase how to translate other modifications for search engines. As pFind already uses UniMod names, AlphaBase supports all of pFind modifications, thus parsing open-search results is

straightforward.

pp 13, lines 409-410 : “We further used AlphaPeptDeep to inspect retention time and MS2 spectrum similarities (Extended Fig. 8)”

You need to say something about the conclusions you drew from this analysis, especially since the figure is not in the main text.

Reply: We agree and updated the text in the manuscript:

“The results demonstrated an 80% PCC90 of phospho-HLA PSMs which is close to unmodified ones, and RT differences from predicted to measured peptides were also close to zero (Extended Fig. 8). Furthermore, we manually validated 300 randomly selected HLA PSMs for these HLA peptides including 44 phosphopeptides using an extension to AlphaViz. Their annotated and mirrored MS2 plots can be found in Suppl. Data 4. This independently verified our model and assignments.”

pp 13, lines 410-412 : “Note that the MS2 and RT models were only fine-tuned by at most 100 phospho-PSMs from eight RAW files (Online Methods), so most of the phosphopeptides from other RAW files were not used in fine-tuning.”

What other RAW files do you mean? I don't quite follow here.

Reply: To make this clear, we now mention the RAW file number when introducing the HLA DDA datasets: “To investigate how much AlphaPeptDeep can boost the HLA peptide search, we applied it on two datasets, MSV000084172 containing 424 RAW files from samples in which particular mono-allelic HLA-I types were enriched, here referred to as the ‘mono-allelic dataset’ and our published dataset from tumor samples (PXD00489444 with 138 RAW files) referred to as the ‘tumor dataset’.”

Having said this, we can explain other RAW files better in the revised manuscript:

“Note that the MS2 and RT models were only fine-tuned by at most 100 phospho-PSMs from eight RAW files (Online Methods), so most of the phosphopeptides from remaining RAW files (i.e. 416 out of 424 and 130 out of 138 RAW files in the mono-allelic and tumor dataset, respectively) were not used in fine-tuning.”

pp 13, lines 412-414 : “Our method was also able to identify other PTMs associated with HLA peptides, such as cysteinylolation”

In the open-search pFind results when you applied AlphaPeptDeep, was the improvement for modified and unmodified peptides proportionally similar? Or did it result in a larger improvement for modified or for unmodified peptides?

Reply: This is a very good question relating to the quality of our PTM prediction. To answer this, we compared AlphaPeptDeep results with pFind for all HLA peptides, those with regular modifications and those with unexpected PTMs. The results are displayed in the updated Fig. 5b below and in

the revised manuscript. This analysis reveals that, for overall numbers, the improvement is mostly on regular peptides. But the percentage of improvement for unexpected PTMs is much higher.

We also updated the figure caption accordingly: “(a) Improvement upon regular search (MaxQuant). IAA refers to iodoacetamide alkylated peptides. (b) Improvement upon open-search (pFind). Regular peptides here refer to peptides without modifications or those with only Met-oxidation and Cys-alkylation.”

p 15, lines 437-439 : “So far, these efforts have been restricted to experimental DIA libraries because analysis with a predicted HLA library is far more challenging than with an experimental one.”

I believe the Bassani-Sternberg group published some work comparing different strategies for immunopeptidomics DIA library generation, and one of their libraries was built with Prosit predictions (pmid: 33845167). However, I’m not sure if they used computational methods to come up with the peptide list for this library, or if the peptide list was based on experimental results.

Reply: They built an experimental spectral library called BigLib from different sources of HLA DDA data. They also predicted a library based on peptides from BigLib. Then they compared the performance of DIA search between these two libraries. The experimental BigLib turned out to be still better than the predicted BigLib. For that work, the BigLib still came from experimental DDA runs, so they still could not avoid generating experimental DDA libraries. By ‘a predicted HLA library’, we meant the predicted library from proteome sequences or fasta files, which is the challenge in this case. We modified the text to

“So far, these efforts have been restricted to experimental DDA libraries because analysis with a predicted HLA library from proteome sequences is far more challenging than with an experimental one”.

p 15, lines 460-461 : “In brief, we first trained a pan-HLA prediction model with peptides from all known HLA types”

I assume this would be a binary prediction model: 1 for HLA-associated peptide, 0 for not. What

did you use for peptide sequences which were labeled with “0”? Were they random? Biological in origin, but not appearing in any HLA peptide database? Or did the databases already include “not HLA” peptides?

Reply: Yes, this is a binary model, as mentioned in the manuscript. “This model - a binary LSTM classifier predicts if a given sequence is likely to be an HLA peptide presented to the immune system and extracts these peptides from the human proteome sequence.”

For negative samples (peptides): these are randomly selected from proteome sequences with the same length distribution as the HLA peptides. These “0” peptides are not expected to have biological meaning.

We have updated the following text to the revised manuscript in Online Methods:

“Negative samples were randomly picked from the human protein sequences with the same length distribution as the HLA peptides. These samples were then split 80% for training and 20% for testing.”

p 15, line 461 : “all known HLA types”

There are approximately 25,000 named HLA alleles, but you later indicate that the model is trained on 94 HLA types (line 474). So it should just say it is trained on data from 94 HLA types.

Reply: Here, as of a pipeline, we would like to make it more generic for users or readers, meaning that users can train the pan-HLA model based on all of their HLA types of interest. We removed “all” from the sentence.

p 15, lines 461-462 : “Normally, only a few HLA types are actually present in the samples from any given individual”

Up to 6 HLA alleles per individual

Reply: Thanks, we updated the text to: “Normally, up to 6 different HLA types are present in the samples from any given individual.”

p 16, line 475 : “with 82% sensitivity.”

How did you calculate this sensitivity? Is it based on the direct-DIA results?

Reply: We split the peptides from 94 HLA types into training (80%) and testing (20%) sets. The sensitivity is from the test set. We now describe this as: “We started with our pan-HLA prediction model trained by 80% of the peptides and tested on the remaining 20% from the 94 known HLA types (Fig. 5). This reduced the number of sequences from 70M to 7M with 82% sensitivity on the testing set.”

p 16, lines 484-486 : “The number of HLA peptides predicted by this model is 3M, which is comparable to the tryptic human proteome library.”

I would say that both 3M and 7M are comparable to 900,000 when compared against the original non-specific peptide space of 70M. Perhaps better to say that 3M is more comparable.

Reply: Yes, fixed as “The number of HLA peptides predicted by this model is 3M, which is more comparable to the tryptic human proteome library.”

p 16, line 504-505 : “To simulate this situation, we removed all peptides of the dominant HLA-A*68:01 and used the rest to train a new pan-HLA model”

It is not explicitly clear here what from what dataset you are removing these peptides. I assume you mean the dataset that was used to train the pan-allele model?

Reply: Yes, we rephrased it as “To simulate this situation, we removed all peptides of HLA-A*68:01 allele from the 94 HLA allele types, and used the rest to train a new pan-HLA model”

How did you choose this particular HLA type to remove? It would seem best to remove an allele whose binding motif is distinct from all the other alleles in the training set, but this can be hard to do because there is a lot of overlap in the binding specificity of HLA alleles. Just looking through the first several alleles in the 94-allele dataset and comparing their binding motifs with A*68:01, I saw that A*11:01, A*11:02, A*33:01, A*33:03 and A*34:02 were similar at the anchor residues. There could be many more, so even though your allele is absent from the training set, the pan-HLA model might still recognize the peptides as HLA peptides. In general, this is a good thing, I think! But it means it might be difficult test the pipeline on truly “unseen” HLA alleles by removing a single allele from the dataset.

Possibly a better option would be test your pipeline on data from something non-human, like a mouse dataset. There are good predictors for mouse HLA presentation, so you should still be able to get a good idea about the sensitivity of the library you build. If you can find a DIA dataset for mouse, that is... I think maybe something came out of the Abersold group back in the early days of SWATH. There is the mouse atlas, but I don't recall if that has any DIA data in it or not.

Another possibility would be to go through all 94 alleles and look at their motifs (DTU Health Tech has a good motif viewer (look for the “ Motif viewer ” tab): <https://services.healthtech.dtu.dk/service.php?NetMHCpan-4.1>). You could remove every allele that resembles A*68:01. Or pick an allele with something really easy to spot, like the giant E at position 1 of B*40:01, and then remove everything that looks like that.

Reply: This is a very good question, we think the key concern here is that even if we removed A6801 peptides from the pre-training data, other similar HLA types may still provide the possibilities to predict A6801 peptides after transfer learning. We agree that using mouse MHC peptides would a good idea, but we have asked around including in a conference about immune systems, and it appears that there are no public mouse MHC DIA yet available yet.

Following the reviewer’s “removing every allele that resembles A6801” idea, alternatively, we thought that we could remove all allele types, meaning that we train the sample-specific model without pre-trained models, such that all allele types are now “unknown”. The results without pre-training are shown in figure below and in the revised Fig. 6b. We also updated the text accordingly:

“Interestingly, training directly on the direct-DIA peptides without transfer learning resulted in only slightly fewer peptides than transfer learning (“no pretrain” in Fig. 6b). As there are not many motifs for each individual, we reasoned that it should be straightforward to learn the sequential patterns by only thousands of HLA peptides. This would be useful to identify unknown sample-specific HLA allele types as we do not need any prior knowledge. Note that transfer learning is still necessary if we only have small number of training peptides and unknown allele types. To simulate this situation, we removed all peptides of HLA-A*68:01 from the 94 allele types, and used the rest to train a new pan-HLA model.”

pp 21, lines 679-680 : “Model fine-tuning. 5,000 PSMs are randomly sampled from at most eight RAW files at 1% FDR to fine-tune the MS2, RT and CCS”

What metric is used for the FDR calculation? Does it depend upon FDR having been previously calculated by a tool, or can you specify a metric to use?

Reply: Yes, it is based on search engine FDR. We have updated the statement.

“Model fine-tuning. To reduce overfitting, only 5,000 PSMs are randomly sampled from at most eight RAW files at 1% original FDR reported by the search engine to fine-tune the MS2, RT and CCS (if applicable) models to obtain project-specific models.”

p 21, lines 688-689 : “61 score features”

In the supplemental data I see that there is only one feature for “Search engine score”. It is

common for search engines to report multiple scores. Does AlphaPeptDeep choose a single score to include? Or do they all end up in the Percolator features?

Reply: "Search engine score" is the primary score to calculate FDR in search engines, we fixed this in suppl. data 2.

We could actually support multiple scores, see `other_score_column_mapping` in https://github.com/MannLabs/alphapeptdeep/blob/main/peptdeep/constants/default_settings.yaml#L172. We currently only support for pFind (Final_Score and Raw_Score) and MSFragger (hyperscore and nextscore in its pepxml), but we can configure the yaml settings file to support multiple scores for other search engines. However, we believe that these features are too low-level for general audiences.

Reviewer #3 (Remarks to the Author):

The manuscript first describes a deep learning framework that can learn and predict peptide properties - retention time, collisional cross sections and fragment ion intensities possibly for modified peptides. It also facilitates adaptation of a pre-trained general model to different tasks that involve the prediction of various peptide properties. The latter part of the manuscript describes how this framework can be used to set up a data analysis pipeline for HLA peptides and improve HLA peptide identification.

There is nothing new about AlphaPeptDeep itself. Almost all the main components have already been published: Deep learning has been applied and proven to be effective in predicting MS2 spectra and retention time only from peptide sequences; Transfer learning has been shown to be effective in predicting MS2 spectra of modified peptides, given a trained model for unmodified peptides; Rescoring peptide-spectrum matches using predicted MS2 spectra has been shown to improve peptide identification performance. While offering a user-friendly interface for transfer learning can be helpful, it is not worth another publication.

Reply: We acknowledge that deep learning has already been applied to the problems that the reviewer mentions (though not to some that we are also tackling in the manuscript). With due respect, we disagree that only the first paper on any topic like this can contain any novelty, i.e. there are thousands of useful papers on computer vision with different or similar methods.

However, there are some intriguing aspects about its application to an HLA peptide analysis pipeline. AlphaPeptDeep was used to create a pan-HLA prediction model in the first place, and transfer learning was then adopted to classify and select HLA peptide sequences specific to particular samples, reducing the number of sequences required to generate a library of predicted spectra and eliminating the need for a DDA experiment to build a spectral library for DIA search.

Reply: We thank the reviewer for the positive comments about our analysis of HLA peptidomics.

While this approach to DIA-HLA analysis is interesting, the authors did not make it clear why the DIA experiment is superior to DDA in HLA peptide analyses. The primary goal of

immunopeptidomics is the discovery of HLA peptide sequences. There are pros and cons for both the DDA and DIA experiments, and it has been and remains unclear which of the two gives better performance in peptide identification, if quantitation is not an issue. The DIA experiment is an overarching presumption in the proposed HLA peptide analysis pipeline, because DIA data analysis requires the construction of a spectral library, and its size needs to be controlled for various reasons (such as the nonspecific digestion of HLA peptides, FDR control, etc.), necessitating the need for predicting HLA peptides that are known to be sample specific.

Reply: We did not specifically mean to make the case that DIA is better than DDA for HLA analysis and we agree that this would require more proof than we offer in this manuscript. (That said, based on our recent results, we would expect this to be the case.) Here we just wanted to demonstrate that our DIA pipeline can improve on existing DDA and DIA workflows.

If the DDA results were searched in our predicted HLA peptides, DDA would likely identify more peptides. However, as the current DDA search engines used here do not support library search, we did not compare this in our work.

We discussed this some more in the Conclusion section of the revised manuscript:

“Using our predicted libraries outperformed existing DDA and DIA workflows. However, this does not prove that DIA is always better than DDA in HLA peptidome analysis, as the DDA proteome database is 20 times larger than our predicted library for DIA analysis. Future DDA search engines may be able to identify more peptides if they supported predicted library search.”

It is unclear how "DDA" was obtained when comparing the number of identified HLA peptides (Fig. 6b). In Fig. 5a, the authors demonstrated that the MaxQuant search with rescoring produced significantly improved outcomes. Therefore, "DDA" number in Fig. 6b would be a lot more informative if the rescoring was applied.

Reply: Thank you for this suggestion. We have now used pFind with AlphaPeptDeep rescoring which resulted in the largest number of HLA sequences for DDA data, see the figure below (grey bar). Now the DDA number is slightly better than PEAKS-Online, but DIA with the AlphaPeptDeep library still achieved the most HLA peptide identifications.

The corresponding text in the revised manuscript is updated as “As a reference, the DDA search on the data in the original publication but using Open-pFind rescored by AlphaPeptDeep identified 30,039 unique sequences in the 8 to 14 aa range”.

It is my opinion that the authors restructure the manuscript by focusing on the proposed HLA peptide analysis pipeline and its performance gain. It can certainly be added that AlphaPeptDeep, which offers a user-friendly interface to deep learning and transfer learning in the prediction of peptide properties, was very helpful in the process of building the pipeline.

Reply: We believe that the lasting contribution of AlphaPeptDeep will be more as a framework than just a model for a very important but still quite specialized application. Therefore, we would like to cover different prediction problems for different audiences from different areas. Our results demonstrate that AlphaPeptDeep achieves state of the art results for different and even novel problems. All of these advances come together in the challenging problem of open-search rescoring in the HLA-DDA section, and HLA peptide and library prediction in the HLA-DIA section. We believed all these different applications make our manuscript more complete and useful. We also communicated with the editor on this point, who is of the same opinion.

REVIEWERS' COMMENTS

Reviewer #1 (Remarks to the Author):

This revised manuscript addressed my previous concern by manually validating some additionally identified HLA peptides. The results look promising.

(In the point-by-point response) In the last line of page 1, 90% PCC90 was achieved for unmodified and phosphorylated HLA peptides, but 80% PCC90 was mentioned later in line 5 of page 2. Are they supposed to be the same?

(Line 195) How many precursors are there in the spectral library predicted for the human proteome?

(Line 445) What is SP?

Reviewer #2 (Remarks to the Author):

I am impressed by the documentation and examples the authors have created for the tool. The package feels much more approachable now. Further, they have adequately addressed all of my concerns regarding the manuscript.

Point by point response to reviewers of “AlphaPeptDeep: A modular deep learning framework to predict peptide properties for proteomics”

Reviewer #1 (Remarks to the Author):

This revised manuscript addressed my previous concern by manually validating some additionally identified HLA peptides. The results look promising.

(In the point-by-point response) In the last line of page 1, 90% PCC90 was achieved for unmodified and phosphorylated HLA peptides, but 80% PCC90 was mentioned later in line 5 of page 2. Are they supposed to be the same?

Reply: They are similar but a bit different. 80% PCC90 was calculated on all the identified 359 phospho-HLA peptides in Extended Fig. 8; while 90% PCC90 is calculated on the 44 randomly selected and plotted phospho-HLA peptides in Suppl. Data 4. These 44 phospho-HLA peptides are included in the 359 phospho-HLA peptides.

(Line 195) How many precursors are there in the spectral library predicted for the human proteome?

Reply: The number has been shown in Supplementary Fig. 3b. Now we also added the numbers in the revised manuscript:

“Using these pre-trained models and specifically designed data structures (Online Methods), the prediction of a spectral library with MS2 intensities, RT, and ion mobilities (converted from CCS, Online Methods) for the human proteome with 2.6M peptides and 7.9M precursors took only 10 min on a regular GPU and 100 minutes on the CPU with multiprocessing (Supplementary Figure 3).”

(Line 445) What is SP?

Reply: SP refers to Ser-Pro, we now use “Ser-Pro” in the revised manuscript.